Resource

# Small RNA-Seq reveals novel miRNAs shaping the transcriptomic identity of rat brain structures

Anaïs Soula[1,2], Mélissa Valere[1,2], María-José López-González[1,2], Vicky Ury-Thiery[1,2], Alexis Groppi[3], Marc Landry[1,2] ⓘ, Macha Nikolski[3,4], Alexandre Favereaux[1,2] ⓘ

In the central nervous system (CNS), miRNAs are involved in key functions, such as neurogenesis and synaptic plasticity. Moreover, they are essential to define specific transcriptomes in tissues and cells. However, few studies were performed to determine the miRNome of the different structures of the rat CNS, although a major model in neuroscience. Here, we determined by small RNA-Seq, the miRNome of the olfactory bulb, the hippocampus, the cortex, the striatum, and the spinal cord and showed the expression of 365 known miRNAs and 90 novel miRNAs. Differential expression analysis showed that several miRNAs were specifically enriched/depleted in these CNS structures. Transcriptome analysis by mRNA-Seq and correlation based on miRNA target predictions suggest that the specifically enriched/depleted miRNAs have a strong impact on the transcriptomic identity of the CNS structures. Altogether, these results suggest the critical role played by these enriched/depleted miRNAs, in particular the novel miRNAs, in the functional identities of CNS structures.

## Introduction

miRNAs comprise one of the most abundant classes of gene regulatory molecules in the organism. They are small (22 nt) endogenous noncoding RNAs operating a negative translational regulation on mRNAs (Bartel, 2004). miRNAs are involved in the regulation of most signaling pathways, both in physiology and in pathology, in a broad range of organisms from viruses to animals (Carrington & Ambros, 2003). Their biogenesis mostly starts by the transcription of the miRNA gene by RNA pol II, generating an imperfect stem-loop structure, called the pri-miRNA (Bartel, 2004; Cai et al, 2004; Lee et al, 2004). The pri-miRNA is further processed several times and transported into the cytoplasm. Briefly, the enzyme Drosha cleaves the single-stranded primary miRNA transcripts to produce a stem-loop secondary structure: the pre-miRNA.

Then, the pre-miRNA is cleaved by the enzyme Dicer to produce a mature miRNA duplex (Bernstein et al, 2001). Finally, one of the two strands of the mature miRNA duplex, called the guide strand, is incorporated into a proteic complex called the RNA-induced silencing complex (RISC) complex. From this point, this miRNA-RISC complex, referred to as the miRISC, will exert a translation inhibition on target mRNA (Lee et al, 2002). The target recognition by the miRNA is mainly based on a Watson–Crick pairing between a specific region of the miRNA called the seed region (nucleotides two to eight) and a complementary sequence on the 3′UTR of the target mRNA (Bartel, 2009; Wang, 2014). In addition to the seed region binding, the remainder of the miRNA can be engaged in an imperfect or perfect base pairing with the target mRNA, resulting respectively in translational inhibition and/or degradation of the target mRNAs. Eventually, both interaction modes induce a decrease in target mRNA translation into protein. Because of the imperfect binding between miRNAs and target mRNAs, miRNAs are able to precisely regulate the expression of dozens, if not hundreds, of genes. Indeed, a single miRNA can bind to numerous different mRNAs, whereas a single mRNA can present binding sites for multiple miRNA species (Friedman et al, 2009). Consequently, miRNAs are considered key players in the regulation of cellular gene expression.

The importance of miRNAs in the regulation of the main brain functions is now well established. Thus, miRNAs are involved in many important physiological processes, such as the development and the maturation of the nervous system (Díaz et al, 2014), synaptic plasticity, learning, and memory (Bredy et al, 2011; Aksoy-Aksel et al, 2014; Letellier et al, 2014; Rajman et al, 2017). In addition, miRNAs are involved in numerous neurodegenerative diseases, such as Alzheimer (Sarkar et al, 2016), Parkinson, Huntington, and amyotrophic lateral sclerosis (Nelson et al, 2008; Maciotta et al, 2013; Tan et al, 2015).

For example, the role of two brain-specific miRNAs, miR-134 and miR-124, has been well described. miR-134 regulates the size of dendritic spines in rat hippocampal neurons by the translational repression of the Lim-domain-containing protein kinase 1 (LimK1), an activator of actin polymerization (Schratt et al, 2006; Bernard,

[1]University of Bordeaux, Bordeaux, France  [2]Centre Nationale de la Recherche Scientifique (CNRS), Unité Mixte de Recherche 5297, Interdisciplinary Institute of Neuroscience, Bordeaux, France  [3]Centre de Bioinformatique de Bordeaux, University of Bordeaux, Bordeaux, France  [4]CNRS/Laboratoire Bordelais de Recherche en Informatique, University of Bordeaux, Talence, France

Correspondence: alexandre.favereaux@u-bordeaux.fr

2007). Thus, miR-134 regulates the morphology of most excitatory synapses with an impact on synaptic plasticity–dependent functions such as long-term potentiation and memory formation (Gao et al, 2010). miR-124, another brain-specific miRNA, plays a critical role in the transition of progenitor neuronal cells to adult neurons by inhibiting signaling pathways of non-neuronal genes, thereby facilitating the expression of the neuronal transcriptomic identity (Conaco et al, 2006; Makeyev et al, 2007).

Interestingly, miRNAs show different roles depending on the region in which they are expressed. For example, miRNAs are particularly studied in the striatum in the context of the addiction to drugs or alcohol (Bali & Kenny, 2013) because this region serves as a key area for habit learning, one of the major mechanisms for addiction behavior (Volkow et al, 2006). In the spinal cord, miRNAs are mainly studied in a context of spinal cord injury and pain (Favereaux et al, 2011; Elramah et al, 2014; López-González et al, 2017). In the olfactory bulb, in which the regeneration of neurons occurs frequently, the role of miRNAs in neurogenesis mechanisms has been particularly studied. It has been demonstrated that miR-132 plays a critical role to form the basis of a structural plasticity program (Pathania et al, 2012). Another work shows that miR-124 target genes are involved in morphogenesis and neuron differentiation in the olfactory bulb (Li & Ling, 2017).

However, analysis of miRBase (Griffiths-Jones, 2006) (http://www.mirbase.org), the most common database referencing miRNAs, shows a large difference between the number of miRNAs referenced in the mouse (*Mus musculus*, 1,915 mature miRNAs and 1193 precursors), the human (*Homo sapiens*, 2,558 mature miRNAs and 1,881 precursors), and in the rat (*Rattus norvegicus*, 765 mature miRNAs and 495 precursors). Thus, it seems that a large percentage of the miRNAs expressed in the rat are not yet uncovered. Rat is a gold standard model in neuroscience, and it is thus of prime importance to get an extensive knowledge of miRNA expression in the central nervous system (CNS).

In this study, we used the small RNA-Seq technology to perform an exhaustive analysis of the miRNome of the rat CNS. Even if previous studies explored the miRNome of the rat brain, either they focused their attention on one specific structure, such as the hippocampus (Shinohara et al, 2011), and the hypothalamus (Amar et al, 2012), or they analyzed the miRNome of a structure over different developmental stages (Yao et al, 2012; Yin et al, 2015). A more extensive study has been performed on five different regions of the brain, but unfortunately using the microarray technology, which limits the number of miRNA quantified and occludes the discovery of novel miRNAs (Olsen et al, 2009). To our knowledge, an extensive analysis of the miRNome of several structures of rat CNS has never been performed.

We focused our attention on the olfactory bulb, the cortex, the hippocampus, the striatum, and the spinal cord and found that of the 495 miRNAs referenced in miRBase, 365 are expressed in these five CNS structures. In addition, we discovered 90 novel miRNAs, some of them having orthologous sequences in the mouse or human that play a crucial role in the regulation of the functions of neurons. We also showed that each CNS structure expresses a particular miRNome, and interestingly, the novel miRNAs seem to play a key role in defining structure-specific miRNomes. In addition, we quantified the expression of all mRNAs using mRNA-Seq on the very same samples used for miRNA quantification. Then, we correlated miRNA and mRNA

expression data using target predictions and selected the top 20 regulated mRNAs for further analysis. Gene Ontology (GO) (Ashburner et al, 2000; The Gene Ontology Consortium, 2017) analysis revealed that most of the regulated targets were already known for their role in the development or the function of the nervous system.

# Results

## Small RNA sequencing of five structures of the CNS

We performed small RNA-Seq to reveal the miRNome of five structures of the CNS: the olfactory bulb, the cortex, the hippocampus, the striatum, and the spinal cord. To perform a statistical analysis of the miRNome, we dissected these structures in triplicate from three male adult Wistar rats. Thus, we synthesized 15 independent libraries that were finally pooled for sequencing. RNA sequencing performed well, and we obtained a total of 454,577,868 reads from which 1,150,734 (0.3% of total) were filtered out because they were too short (size < 15 nt). To assess the quality of the reads, we used FastQC software (version 0.11.5, default parameters), and all sequences were flagged as good quality (Table S1).

## The CNS expresses known and novel miRNAs

We first checked the number of miRNAs known and referenced in miRBase that were expressed among the five CNS structures. From the 495 miRNAs referenced in rat miRBase, 365 miRNAs were expressed in our samples, corresponding to 73.7% of the miRBase-referenced miRNAs (Fig 1A).

To discover novel miRNAs expressed in the brain, we used the miRPro software (Shi et al, 2015). Based on miRDeep2, miRPro analyzes the sequences obtained from small RNA-Seq experiments seeking for novel miRNAs using the following pipeline. First, miRPro aligns sequences of potential novel mature miRNAs on the genome. Then, the algorithm excises the corresponding potential pre-miRNAs from the genomic sequence. Finally, miRPro determines the most relevant pre-miRNA, in particular by the analysis of pre-miRNA folding with the RNAfold tool (Hofacker, 2003). miRPro analysis of the reads obtained from the five CNS structures leaded to the prediction of 8,416 novel pre-miRNAs (Table S2). To discard potential false-predictive novel pre-miRNAs, we refined the analysis by only conserving those expressed at least in two replicated samples from the same CNS structure and with a minimum expression of 1 count per million (cpm). Finally, we identified 107 novel pre-miRNAs (Table S3), which after miRNA processing gave 90 novel mature miRNAs (Table S4), not yet referenced in rat miRBase. The number of pre-miRNAs was higher than the number of mature miRNAs because 8 novel pre-miRNAs were duplicated in the genome. For example, novel-pre-miR-19 was present on both chromosomes 3 and 20, novel-miR-8 was present in 3 copies on chromosome 1, and novel-miR-2 was present on chromosomes 7, 12, 15, and 18 and twice on chromosome 1 (Table 1).

The length of the novel miRNAs ranged from 18 to 24 nt, with a mean length of 20.59 nt, which is in line with the conventional characteristics of miRNAs. To assess whether these novel miRNAs display the canonical stem-loop structure of miRNAs, we checked the

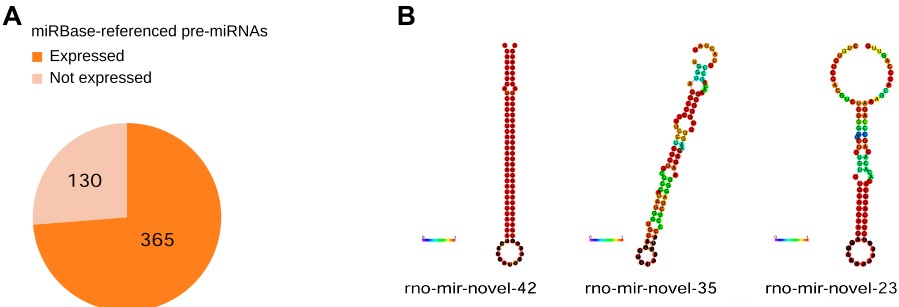

**A** miRBase-referenced pre-miRNAs
▮ Expressed
▮ Not expressed

**B**

rno-mir-novel-42   rno-mir-novel-35   rno-mir-novel-23

**Figure 1. Pre-miRNAs expressed in the CNS structures as revealed by RNA-Seq.**
**(A)** Pie chart of miRBase-referenced pre-miRNAs, 365 of them are expressed in the CNS. **(B)** Predictive structures of novel miRNAs were calculated with the RNAfold program. Colors represent the probability of matching between the bases. The predictive structures of novel miRNAs are similar to those of known miRNAs. **(C)** Histogram of the number of known and novel precursors of miRNAs expressed in each structure. **(D)** Venn diagramm of the repartition of pre-miRNAs in the CNS structures considering known and novel pre-miRNAs.

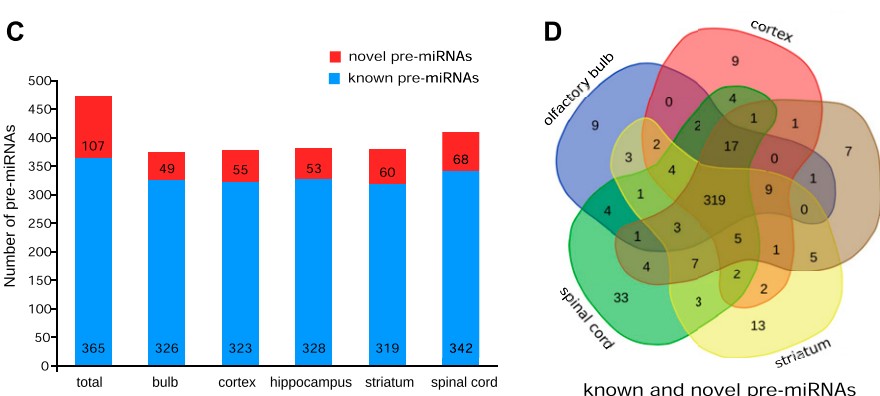

**C** novel pre-miRNAs ▮  known pre-miRNAs ▮

**D**

known and novel pre-miRNAs

predictive folding of the novel pre-miRNA sequences. Fig 1B shows, as an example, the secondary structure of three novel pre-miRNAs: rno-mir-novel-42, the most expressed in our samples; rno-mir-novel-35, an miRNA with an expression level corresponding to the average expression of all novel miRNAs; and finally rno-mir-novel-23, which had the lowest expression. All these novel miRNAs display a stem-loop structure similar to the one of prototypical miRNAs (e.g., rno-mir-124-1, Fig S1). Thus, the predictive secondary structure of these novel pre-miRNAs is compatible with functional miRNAs.

### Pre-miRNA expression in the five CNS structures

We compared pre-miRNA expression across the five CNS structures. Globally, all the studied structures expressed more or approximately the same amount of known pre-miRNA species (from 319 for the striatum and up to 342 for the spinal cord, Fig 1C). In addition, concerning novel pre-miRNAs, four of the five CNS structures under study expressed approximately the same amount of novel pre-miRNAs (49 for the olfactory bulb, 55 for the cortex, 53 for the hippocampus, and 60 for the striatum). In contrast, the spinal cord expressed a higher number of novel pre-miRNAs compared with the other structures, with 68 novel pre-miRNAs, which is more than the half of the total number of novel pre-miRNAs discovered.

Then, we analyzed the repertoire of pre-miRNAs detected in each CNS structure. We used a Venn diagramm representation in which each CNS structure is represented by a color: the cortex in red, the hippocampus in brown, the striatum in yellow, the spinal cord in green, and the olfactory bulb in blue. This representation (Fig 1D)

**Table 1. Chromosomic organization of the novel miRNAs.**

| Name of the novel miRNA | Number of copies | Chromosomic location | Number of chromosomes |
|---|---|---|---|
| rno-novel-miR-2-5p | 6 | 2 x chr 1; chr 7; chr 12; chr 15; chr 18 | 5 |
| rno-novel-miR-8-3p | 3 | 3 x chr 1 | 1 |
| rno-novel-miR-12-5p | 3 | chr 2; 2 x chr 4 | 2 |
| rno-novel-miR-19-5p | 2 | chr 3; chr 20 | 2 |
| rno-novel-miR-22-5p | 5 | chr 3; 2 x chr 5, chr 6; chr 7 | 4 |
| rno-novel-miR-26-3p | 2 | chr 18; chr 5 | 2 |
| rno-novel-miR-52-3p | 2 | chr 10; chr 11 | 2 |
| rno-novel-miR-66-5p | 2 | chr 12; chr 14 | 2 |

Duplicated miRNAs are localized either on the same chromosome or on different chromosomes.

showed that most of the miRNAs, known or novel, were detected in all the five CNS structures studied (319 of 472, 67.58%). However, some pre-miRNAs were detected only in one structure (9, 9, 7, 13, and 33 for the olfactory bulb, the cortex, the hippocampus, the striatum, and the spinal cord, respectively).

**Chromosomal repartition of pre-miRNAs**

To try to understand why these new miRNAs were not discovered before and to look for possible miRNA clusters, we analyzed the localization of the genes coding for the miRNAs along the 22 chromosomes of the rat genome.

First, we studied the strand of the DNA molecule on which miRNAs were located. Globally, there were more miRNAs, known and novel, along the positive strand compared with the negative strand (Fig 2A). In addition, the proportion of miRNAs located on the positive strand is significantly higher for novel miRNAs than for known miRNAs (Fisher exact test, $P < 0.05$). However, the biological meaning of this bias toward the positive strand for miRNA genes is still unknown.

The chromosomal repartition of the miRNAs expressed in the five CNS structures combining both miRBase-referenced and novel miRNAs (Fig 2B) was similar to the one of miRBase-referenced miRNAs only (Fig S2). Indeed, chromosomes 1, 6, and X were the largest contributors to miRNAs expression in the CNS (49, 65, and 46, respectively, Fig 2B and Table S5) and chromosomes 1, 7, and 10 contained the highest number of novel miRNAs (12, 8, and 15 respectively, Fig 2B and Table S5).

Linear regression and statistical analysis showed a moderate correlation between the size of the chromosomes and the number of the miRNAs genes (Fig 2C, $R^2$ = 0.3814, $P$ = 0.0022). To go further on this analysis, we looked for miRNAs clusters. Because criteria for miRNA clusters are versatile in the literature (Altuvia et al, 2005), we selected three different ranges for miRNA clusters. We considered two miRNA genes as clustered if they were located within a 1-kb, 2-kb, or 5-kb region of the genomic DNA. According to these criteria, none of the novel miRNA genes were clustered to other known or novel miRNA gene (Table S6).

Genome mapping analysis of the novel pre-miRNAs revealed that most of them were located in intergenic regions 62 (57.94%, Fig 2D and E) or introns 30 (28.04%, Fig 2D and F); the remaining were located in the promoter transcription start site: 10 (9.35%); transcription termination site: 10 (9.35%); exon: 1 (0.93%); or 3'UTR: 1 (0.93%).

**Novel mature miRNAs**

Maturation of the pre-miRNA produces two miRNAs corresponding to the 5' and 3' ends of the stem of the hairpin structure. One of the miRNAs, called the "passenger" miRNA, is degraded, whereas the other one, the "guide" or "mature" miRNA, is loaded into the miRISC complex to target mRNAs. To decipher the role of miRNAs in the transcriptomic regulation of CNS, it is thus of prime importance to analyze the expression of mature miRNAs.

First, we analyzed the nucleotidic composition of the novel miRNAs. We found that U is the most representative nucleotide in

the 5' position of the novel miRNA sequences, followed by C and A (Fig 3A).

Then, we wondered if, of the 90 novel miRNAs we discovered, some were orthologs with a mouse and/or a human miRNA already referenced in miRBase. Using Blast software (Altschul et al, 1990), we defined as ortholog a miRNA sequence with a minimum of 95% similarities with a miRNA from another species. We found 12 novel rat miRNAs that were orthologs of mouse and/or human miRNAs (Table 2). More particularly, there were five miRNAs that were orthologs of mouse miRNAs only, whereas the other seven miRNAs were orthologs of both human and mouse miRNAs. For example, novel-miR-7-5p and novel-miR-8-3p were canonical orthologs of mouse miR-344-5p and miR-344h-3p, respectively, whereas novel-miR-14-3p was a canonical ortholog of both mouse and human miR-330-5p (Table 2).

miRNA targets are commonly predicted on the basis of a perfect complementarity between the seed region of the miRNA and the 3'UTR of the target mRNA. Thus, miRNAs sharing the same seed region often share the same targets and are grouped as a family of miRNAs (Ambros et al, 2003). Consequently, in addition to canonical orthologs, we searched for what we called "functional orthologs," which are miRNAs sharing, at least, the same seed region with an miRNA from another species.

We found in total 23 "functional orthologs" either of human or mouse miRNAs: 8 "functional orthologs" of mouse miRNAs, 12 "functional orthologs" of human miRNAs, and three novel rat miRNAs sharing the seed region with both mouse and human miRNAs (Table 3). For example, novel-miR-78-5p was a "functional ortholog" of mouse miR-3070-5p, novel-miR-73-3p was a "functional ortholog" of human miR-765, and novel-miR-84-5p was a "functional ortholog" of both human and mouse miR-134-3p. To evaluate the ability of these "functional orthologs" to repress mRNA expression the same way as miRNAs sharing the same seed region, we designed a proof-of-concept experiment based on a luciferase assay. Novel-rno-miR-21-5p was predicted to be a "functional ortholog" of mouse mmu-miR-676-3p and thus may regulate the same target genes. We searched TargetScan for the predicted targets of mmu-miR-676-3p, and from the top five predicted targets, we selected LCE2D. LCE2D stands for late cornified envelope protein 2D, a protein involved in the development of skin. We selected LCE2D because the interaction with mmu-miR-676-3p relied only on the hybridization of the seed region, and interestingly, novel-rno-miR-21-5p was also predicted to interact with LCE2D only via the seed region. We cloned the 3'UTR sequence of LCE2D in a luciferase reporter plasmid and tested in COS cells the ability of novel-rno-miR-21-5p to regulate LCE2D. As a control miRNA, we used Cel-miR-67, a miRNA from *C. elegans* known to have no target in mammals. Fig 3B shows that mmu-miR-676-3p inhibited the expression of the luciferase reporter by 15% compared with control condition, and more interestingly, novel-rno-miR-21-5p inhibited the reporter gene by 22%. The level of regulation was moderate, but this is a common feature of miRNAs that are known as fine-tuner of gene expression. The key point is that both miRNAs were able to regulate the same target, and this can be considered a clue that "functional orthologs" may regulate the same targets.

In conclusion, among the 90 novel mature miRNAs discovered in the rat CNS by small RNA-Seq, only 35 were orthologs, either

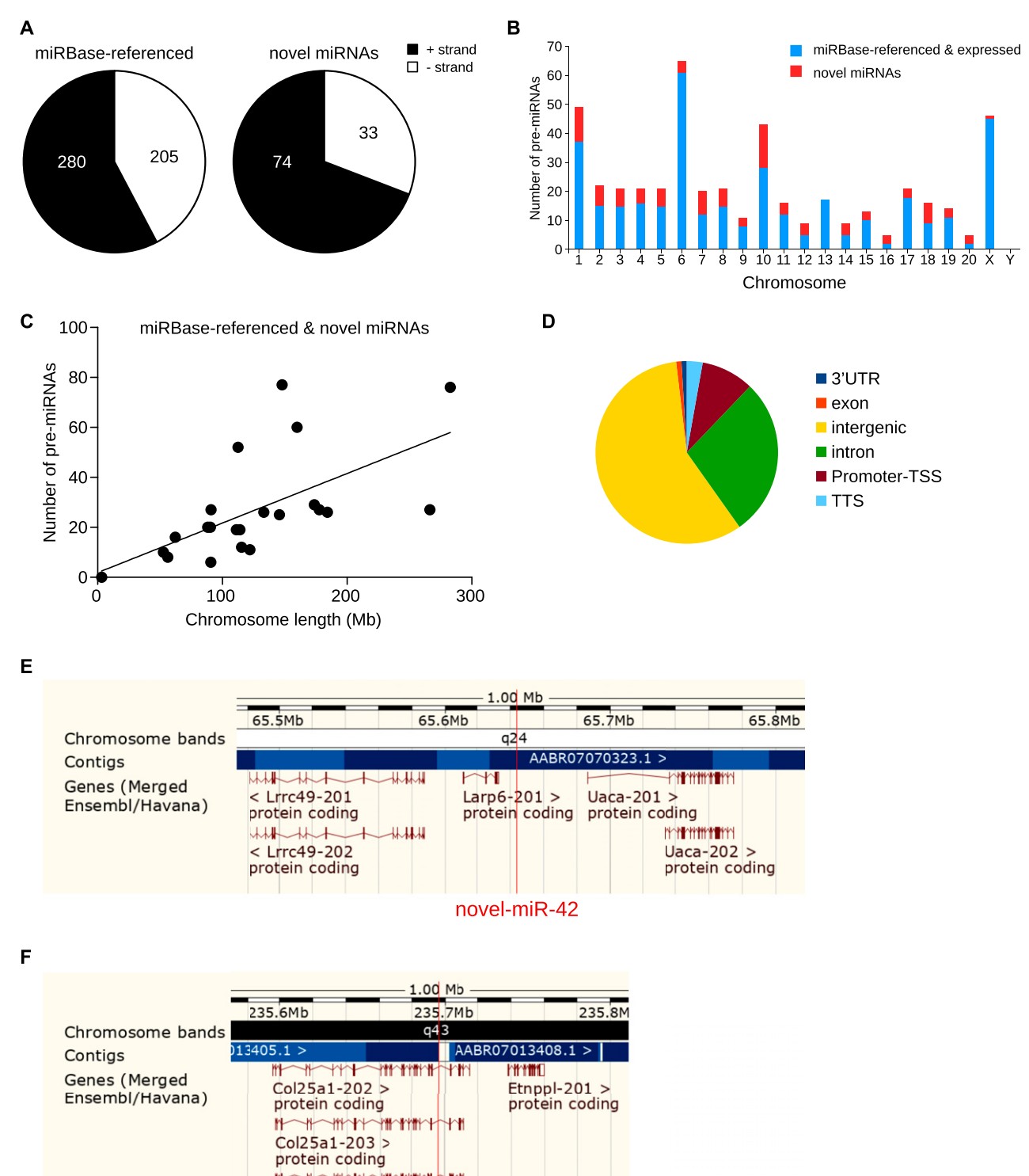

**Figure 2. Chromosomic repartition of miRNAs.**
**(A)** Pie charts of the repartition of pre-miRNAs per DNA strand. There are more novel miRNAs on the positive strand than for miRBase-referenced miRNAs, Fisher exact test, $P < 0.05$. **(B)** Histogram of the repartition of known and novel pre-miRNAs along the 22 chromosomes of the rat genome. **(C)** Correlative analysis of the number of miRNAs in function of the length of chromosomes, $R^2 = 0.3814$, $P = 0.0022$. **(D)** Pie chart represents genome mapping of the novel pre-miRNAs: most of the new miRNAs are intergenic (57.94%) or intronic (28.04%). Genome browser view of an intergenic pre-miRNA (novel-miR-42, **E**) and an intronic pre-miRNA (novel-miR-13, **F**).

**A**

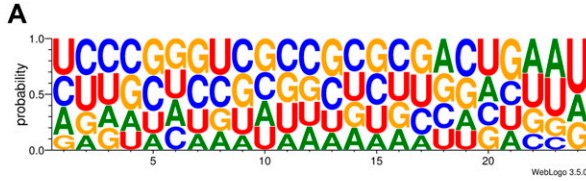

**B**

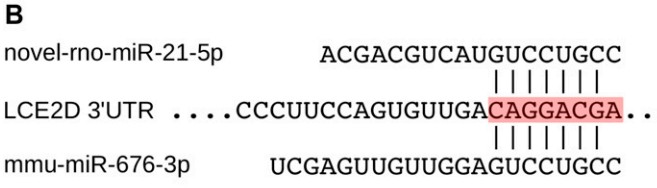

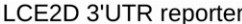

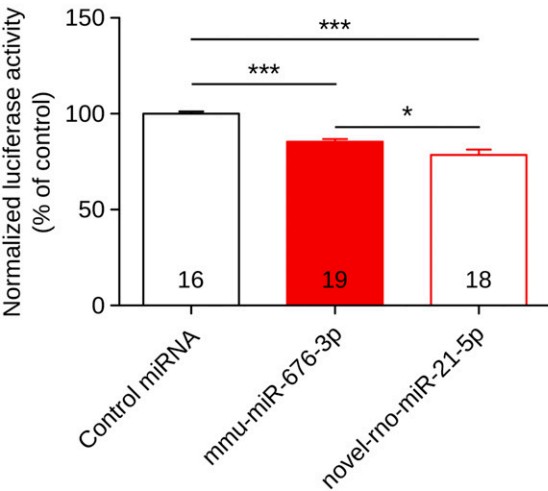

**Figure 3. Analysis of the sequence of novel miRNAs.**
**(A)** The nucleotidic composition of all novel miRNAs was compiled, and the probability of each nucleotide at each position is plotted; U is the predominant nucleotide at the 5′ position. **(B)** Proof of concept for the "functional" ortholog miRNA hypothesis, mmu-miR-676-3p, and novel-rno-miR-21-5p share the same seed region on the LCE2D 3′UTR. Luciferase experiment proves that both miRNAs are able to interact with LCE2D 3′UTR and mediate luciferase translation inhibition. Cel-miR-67, a miRNA from *C. elegans* known to have no target in mammals, is used as control. Data shown are mean ± SD; numbers within the bars indicate biological replicates, one-way ANOVA, followed by the Dunnett post hoc test, ***$P < 0.001$, *$P < 0.05$.

classical or "functional" orthologs, of mouse or human miRNAs. Consequently, most novel miRNAs that we discovered are totally new, with no orthologs at least in the mouse or human.

### Clustering of the CNS structures in regard to their miRNome

To test whether miRNA expression discriminated the different brain structures, we performed hierarchical clustering analysis of the miRNome of the five CNS structures tested. To perform this analysis, we used the count per million value for each miRNA (Table S7), and we represented the results as a heatmap.

### miRbase-referenced miRNAs

First, we analyzed the expression of miRBase-referenced miRNAs in the five CNS structures studied. Hierarchical clustering based on the expression of known miRNAs, followed by heatmap representation, showed that replicates of the same CNS structure are almost all grouped together (Fig 4A). Thus, except for one cortex replicate that was not clustered with the two others biological replicates, all other biological replicates were grouped together. Particularly, we can notice that most of the miRBase-referenced miRNAs were expressed approximately at the same level across the different samples, and this is in line with the Venn diagramm analysis (Fig 1D). However, specific miRNAs were differentially expressed, giving a miRNA signature for each CNS structure. For instance, the high expression level of miR-96-5p, -200a-3p, -200b-3p, and -200c-3p in the olfactory bulb discriminated this structure from the others.

### Novel miRNAs

Then, we did the same analysis on the newly discovered miRNAs. This second heatmap showed a perfect clustering of the three replicates from the same structure (Fig 4B). This perfect clustering may result from the enrichment of some structures for specific miRNAs. For example, novel-miR-47-5p had a low expression in all structures except in the olfactory bulb. The other way around, the low expression of some miRNAs may enable the clustering of a given CNS structure. For instance, novel-miR-28-3p was well expressed in all structures but the cortex. These results suggest that the novel miRNAs discovered may be important to the specification of the CNS structures.

### miRNAs are differentially expressed in CNS structures

Hierarchical clustering analysis suggests that we could use the expression of miRNAs as a criterion for CNS structures classification. To identify which miRNAs were specifying CNS structures, we performed a statistical analysis of miRNA expression data with the DESeq2 algorithm (Love et al, 2014) (default parameters). Thus, we were able to determine miRNAs that were underexpressed or overexpressed in a CNS structure compared with all the others.

We found differentially expressed miRNAs in each structure but in variable amount. A complete list of all differentially expressed miRNAs, their fold changes to the other CNS structures, and statistical *P*-values can be found in Table S8. In particular, the olfactory bulb had the highest number of differentially expressed miRNAs with 44 overexpressed and 34 down-regulated (Fig 5A). In the spinal cord, there were 32 overexpressed miRNAs and 33 downregulated (Fig 5E). Four miRNAs were overexpressed in the cortex and seven down-regulated (Fig 5B). In the hippocampus, only two miRNAs were differentially expressed, one up-regulated and one down-regulated (Fig 5C). Finally, in the striatum, we found 20 miRNAs overexpressed and two underexpressed (Fig 5D).

Interestingly, among the differentially expressed miRNAs, some were newly discovered miRNAs. For instance, in the olfactory bulb, novel miRNAs rno-novel-miR-47-5p and -56-5p were up-regulated, whereas rno-novel-miR-42-5p and -59-3p were down-regulated.

**Table 2. Some novel miRNAs have orthologous sequences in human and/or mouse genomes.**

| Novel rat miRNAs | | Mouse orthologs | | Human orthologs | |
|---|---|---|---|---|---|
| Name | Sequence | Name | Sequence | Name | Sequence |
| rno-novel-miR-7-5p | AGUCAGGCUACUGGUUAUAUUCCA | mmu-miR-344-5p | AGUCAGGCUCCUGGCUAGAUUCCAGG | N/A | N/A |
| rno-novel-miR-8-3p | GGUAUAACCAAAGCCCGACUGA | mmu-miR-344h-3p | GGUAUAACCAAAGCCCGACUGU | N/A | N/A |
| rno-novel-miR-14-3p | UCUCUGGGCCUGUGUCUU | mmu-miR-330-5p | UCUCUGGGCCUGUGUCUUAGGC | hsa-miR-330-5p | UCUCUGGGCCUGUGUCUUAGGC |
| rno-novel-miR-24-5p | UCAAGAGCAAUAACGAAAA | mmu-miR-335-5p | UCAAGAGCAAUAACGAAAAAUGU | hsa-miR-335-5p | UCAAGAGCAAUAACGAAAAAUGU |
| rno-novel-miR-35-5p | UUUCCUCUCUGCCCCAUAGGGU | mmu-miR-3059-5p | UUUCCUCUCUGCCCCAUAGGGU | N/A | N/A |
| rno-novel-miR-37-3p | GCAGGAACUUGUGAGUCU | mmu-miR-873a-5p | GCAGGAACUUGUGAGUCUCCU | hsa-miR-873-5p | GCAGGAACUUGUGAGUCUCCU |
| rno-novel-miR-38-5p | ACUCUAGCUGCCAAAGGCGCU | mmu-miR-1251-5p | ACUCUAGCUGCCAAAGGCGCU | hsa-miR-1251-5p | ACUCUAGCUGCCAAAGGCGCU |
| rno-novel-miR-56-5p | ACUGGACUUGGAGUCAGAAG | mmu-miR-378c | ACUGGACUUGGAGUCAGAAGC | hsa-miR-378a-3p | ACUGGACUUGGAGUCAGAAGGC |
| rno-novel-miR-64-5p | CUAAGGCAGGCAGACUUCAGUGU | mmu-miR-6540-5p | CUAAGGCAGGCAGACUUCAGUG | N/A | N/A |
| rno-novel-miR-72-3p | CCAGUAUUGACUGUGCUGCUGAA | mmu-miR-16-1-3p | CCAGUAUUGACUGUGCUGCUGA | hsa-miR-16-1-3p | CCAGUAUUAACUGUGCUGCUGA |
| rno-novel-miR-87-5p | GUUCCUGCUGAACUGAGCCAGU | mmu-miR-3074-5p | GUUCCUGCUGAACUGAGCCAGU | hsa-miR-3074-5p | GUUCCUGCUGAACUGAGCCAG |
| rno-novel-miR-90-5p | AGGUCCUCAGUAAGUAUUUGUU | mmu-miR-1264-5p | AGGUCCUCAGUAAGUAUUUGUU | N/A | N/A |

Novel-miRNAs with their corresponding orthologous sequences in the mouse and/or human. We defined as ortholog an miRNA sequence with a minimum of 95% similarities with an miRNA from another species.

In total, 9 of the 90 newly discovered miRNAs were differentially expressed in CNS structures. These results suggest that the expression of some miRNAs may regulate specifically gene expression in a given structure.

### Structure-specific miRNAs and the regulation of their predicted target genes

The rationale was that to have a strong impact on gene expression and finally on the physiological function of the different CNS structures, specific miRNAs should be highly regulated in one structure compared with all other ones. In addition to the statistical criterion used to define differentially expressed miRNAs, we used a second criterion for specificity. We first searched for enriched miRNAs, and we selected as specifically enriched miRNAs, those with a log 2 of the fold change (log2FC) greater than four, meaning that the expression of the given miRNA was at least 16 times higher in this structure than in any other structures.

In addition to specifically enriched miRNAs, the specific depletion of miRNAs in a structure could have a strong impact on CNS physiology by unleashing gene expression of specific targets. Thus, in addition to specifically enriched miRNAs, we also looked for specifically depleted miRNAs. We selected as depleted miRNAs, miRNAs which had a log 2 of the fold change lower than −4, corresponding to, at least, a 16 times decrease in miRNA expression.

Enrichment/depletion analysis of the CNS miRNome showed that except for the striatum, all structures had specific miRNAs. In particular, there were many miRNAs enriched in the olfactory bulb (12) and in the spinal cord (6) but none in the cortex and in the hippocampus (Fig 6). Concerning down-regulated miRNAs, five miRNAs were depleted in the spinal cord but only one in the olfactory bulb, the cortex, and the hippocampus. Interestingly, the only specific miRNA of the cortex was a novel miRNAs, rno-novel-miR-28-3p. In addition, of the five specifically depleted miRNAs in the spinal cord, three were novel miRNAs (Fig 6).

These results suggest that these specific miRNAs should differentially modulate the transcriptome of these CNS structures. To test this hypothesis, we quantified the expression of all mRNAs using mRNA-Seq on the very same samples used for miRNA quantification. We first determined regulated mRNAs by performing a statistical analysis of mRNA expression data with the DESeq2 algorithm (Love et al, 2014) (Table S9, default parameters). Then, we correlated miRNA and mRNA expression data using target predictions as determined by TargetScan (default parameters) for known miRNAs (Agarwal et al, 2015) and miRDB tools for novel miRNAs. This correlation analysis revealed many miRNA-mRNA pairs regulated in the CNS structures studied except for the striatum (Table S10). Next, we assessed, for each structure, if predicted target genes of the specific miRNAs were more regulated than nontarget genes. To do so, we analyzed the cumulative

**Table 3. Novel miRNAs with their corresponding functional orthologous sequences in the mouse and/or human.**

| Name | Sequence | Name | Sequence | Name | Sequence |
|---|---|---|---|---|---|
| rno-novel-miR-78-5p | UGCCCCUGCUGAGGCUGUGUCU | mmu-miR-3070-5p | AGCCCCUGACCUUGAACCUGGGAAGCCCC UGACCUUGAACCUGGGA | N/A | N/A |
| rno-novel-miR-79-3p | UUAGGACUCUGGUCAUCUUUGG | N/A | N/A | hsa-miR-3117-3p | AUAGGACUCAUAUAGUGCCAGAUAGGA CUCAUAUAGUGCCAG |
| rno-novel-miR-60-5p | CAGGGAGAGAACUAGUACAG | N/A | N/A | hsa-miR-6760-5p | CAGGGAGAAGGUGGAAGUGCAGACAGGG AGAAGGUGGAAGUGCAGA |
| rno-novel-miR-86-3p | ACAGACUGGUGCCUGGGUGUGG | N/A | N/A | hsa-miR-4494 | CCAGACUGUGGCUGACCAGAGGCCAGA CUGUGGCUGACCAGAGG |
| rno-novel-miR-63-5p | GCCUGAGAGCUGGGGGGUA | N/A | N/A | hsa-miR-4324 | CCCUGAGACCCUAACCUUAACCCUGA GACCCUAACCUUAA |
| rno-novel-miR-21-5p | CCGUCCUGUACUGCAGCA | mmu-miR-676-3p | CCGUCCUGAGGUUGUUGAGCUCCGUCCU GAGGUUGUUGAGCU | hsa-miR-676-3p | CCGUCCUGAGGUUGUUGAGCUCCGU CCUGAGGUUGUUGAGCU |
| rno-novel-miR-54-3p | UGUGUCUUUCUCCCUGUUGAC | N/A | N/A | hsa-miR-4711-3p | CGUGUCUUCUGGCUUGAUCGU GUCUUCUGGCUUGAU |
| rno-novel-miR-57-3p | CUCUCACCUCCCGCUCUCCACA | N/A | N/A | hsa-miR-1229-3p | CUCUCACCACUGCCUCCCCACAGC UCUCACCACUGCCUCCCCACAG |
| rno-novel-miR-48-5p | AUGGCGGCUGGAGUCUG | N/A | N/A | hsa-miR-6770-3p | CUGGCGGCUGUGUCUUCACAGCUG GCGGCUGUGUCUUCACAG |
| rno-novel-miR-84-5p | CUGUGGGCUGCAGGAGGAC | mmu-miR-134-3p | CUGUGGGCCACCUAGUCACCCUGUGG GCCACCUAGUCACC | hsa-miR-134-3p | CUGUGGGCCACCUAGUCACCCUGU GGGCCACCUAGUCACC |
| rno-novel-miR-80-3p | CCCAGGGAGCGUAAAGAGCCG | mmu-miR-7226-5p | GCCAGGGAAGUUGAUUGUGUGAAGGGGCCAGG GAAGUUGAUUGUGUGAAGGG | N/A | N/A |
| rno-novel-miR-59-3p | UCCCCUGGGUCUGUGCUCUGCA | mmu-miR-331-3p | GCCCCUGGGCCUAUCCUAGAAGCCCCUGGG CCUAUCCUAGAA | hsa-miR-331-3p | GCCCCUGGGCCUAUCCUAGAAGCCC CUGGGCCUAUCCUAGAA |
| rno-novel-miR-25-3p | GCGGGCGGCGGGGAGGCG | mmu-miR-5126 | GCGGGCGGGGCCGGGGGCGGGGGGCGGGGCG GGGCCGGGGGCGGGG | N/A | N/A |
| rno-novel-miR-3-3p | AUAAGUGUAGAGAGUCUGUAGU | mmu-miR-668-5p | GUAAGUGUGCCUCCGGUGAGCAUGGUAAGU GUGCCUCGGGUGAGCAUG | N/A | N/A |
| rno-novel-miR-71-5p | CUCCCUUAGUCCUCUUGGUUGU | N/A | N/A | hsa-miR-642a-5p | GUCCCUCUCCAAAUGUGUCUUGGU CCCUCUCCAAAUGUGUCUUG |
| rno-novel-miR-9-3p | CUGGGCGGGAUGGAGGUGG | N/A | N/A | hsa-miR-1228-5p | GUGGGCGGGGCGGGGCAGGUGUGUG |
| rno-novel-miR-17-5p | GCAAGGCCCAGCGAGUGACU | N/A | N/A | hsa-miR-3922-5p | UCAAGGCCAGAGGUCCCACAGCA |
| rno-novel-miR-41-5p | UCCGGGGCUGCGGGAUGA | mmu-miR-673-3p | UCCGGGGCUGAGUUCUGUGCACC | N/A | N/A |
| rno-novel-miR-32-5p | ACUCGGAUCAAGCUGAGAGCCA | mmu-miR-6336 | UCUCGGAUUUAGUAAGAGAUC | N/A | N/A |
| rno-novel-miR-49-5p | ACUCUCUCACUCUGCAUGGUUA | N/A | N/A | hsa-miR-7110-3p | UCUCUCUCCCACUUCCCUGCAG |
| rno-novel-miR-73-3p | UGGAGGAGAGAAAAAGAGA | N/A | N/A | hsa-miR-765 | UGGAGGAGAAGGAAGGAGGAUG |
| rno-novel-miR-77-5p | AGUUUUCUGCCUUUCGCUCUGUGG | mmu-miR-7214-5p | UGUUUUCUGGGUUGGAAUGAGAA | N/A | N/A |
| rno-novel-miR-10-3p | AUUCUUCUUCACGUGGGCUUAGA | mmu-miR-6946-3p | UUUCUUCUCUUCCCUUUCAG | N/A | N/A |

Functional orthologous sequences were determined with a perfect homology into the seed region (italicized sequence).

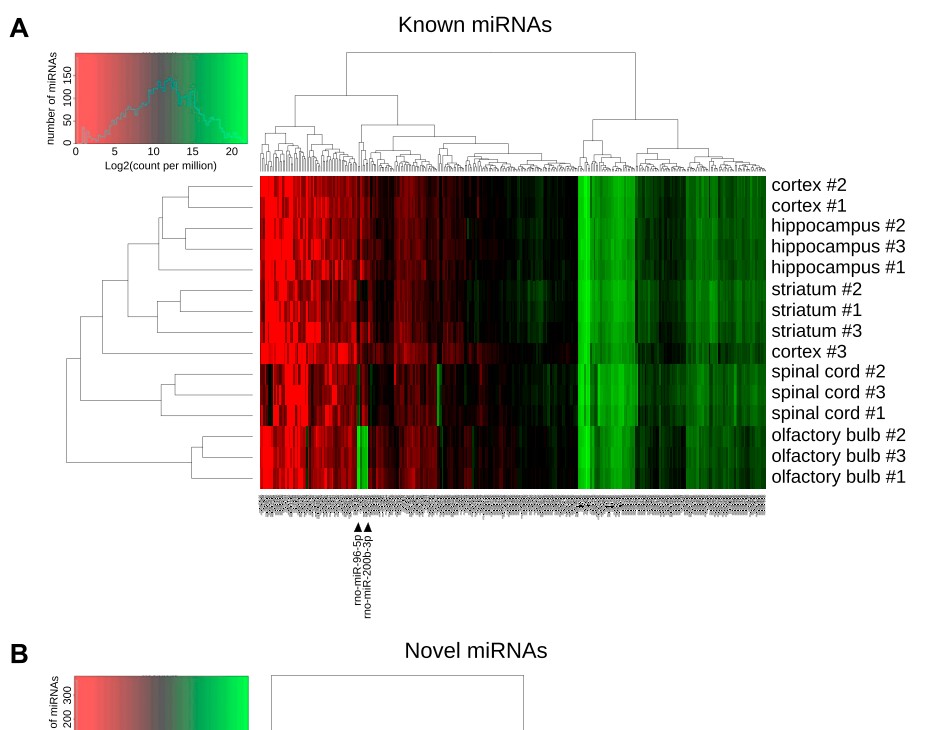

**A**

Known miRNAs

**Figure 4.   Hierarchical clustering of the miRNome of the different CNS structures.**
**(A, B)** Heatmaps representing the expression of each miRNA from all the studied structures. miRNAs are represented at the bottom, biological replicates of the different CNS structures are represented on the right, and statistical dendrogram of clusterization of the samples is represented on the left. Colors represent the level of miRNA expression (log2 of count per million); red: high expression; green: low expression. **(A)** Considering miRBase-referenced miRNAs, all biological replicates of the same structure are grouped together, except cortex replicate #3. **(B)** Hierarchical clustering of novel miRNAs shows a perfect clustering of the biological replicates from the same structure. Black arrowheads indicate miRNAs those expression are strongly different in a structure compared with all other samples.

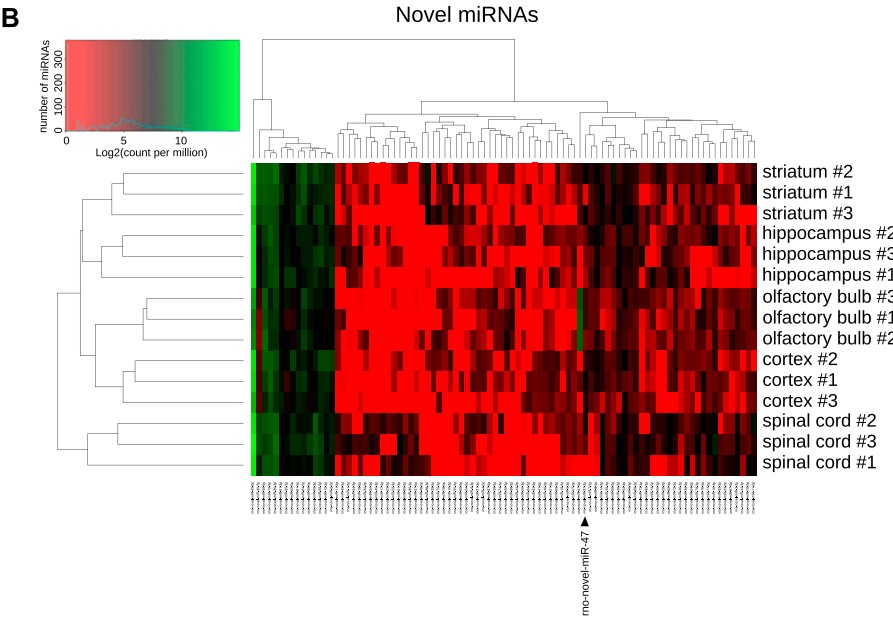

**B**

Novel miRNAs

frequency distribution of expression change of all miRNA target genes versus nontarget genes. Then, in an attempt to highlight the most biologically relevant targets for each structure, we selected the mRNAs displaying the most important regulation (either the 20 uppest or 20 lowest regulated mRNAs). From this list, we performed a GO term enrichment analysis using the Metascape tool (Tripathi et al, 2015). Concerning specifically enriched miRNAs in the olfactory bulb, the cumulative frequency distribution of expression changes of all predicted targets was significantly different from nontarget genes (Fig 7A). Although most of the predicted targets showed an expected down-regulation, some were up-regulated. However, gene-by-gene analysis showed a statistical difference in 12 genes that were all down-regulated (Fig 7B). Interestingly, some regulated mRNAs were predicted to be targeted by multiple structure-specific

miRNAs. For instance, Col25a1 (log2FC = −2.18), a brain-specific membrane collagen, was predicted to be inhibited by miR-96-5p and miR-182. In addition, Caln1 (log2FC = −2.85), a protein with high similarity to the calcium-binding proteins of the calmodulin family, was predicted to be targeted by miR-183-5p and novel-miR-47-5p. Among the most regulated mRNAs were multiple members of the solute carrier family, such as Slc18a3 (log2FC = −4.31) and Slc6a5 (log2FC = −3.48), which transport acetylcholine and glycine, respectively, two important neurotransmitters in the brain. In the same structure, one miRNA was specifically depleted, miR-544-3p (log2FC = −18.6), and the cumulative frequency distribution of expression changes of all its predicted targets was significantly different from nontarget genes (Fig 7C). From the 40 statistically up-regulated targets of miR-544-3p (Table S10), the 20 most

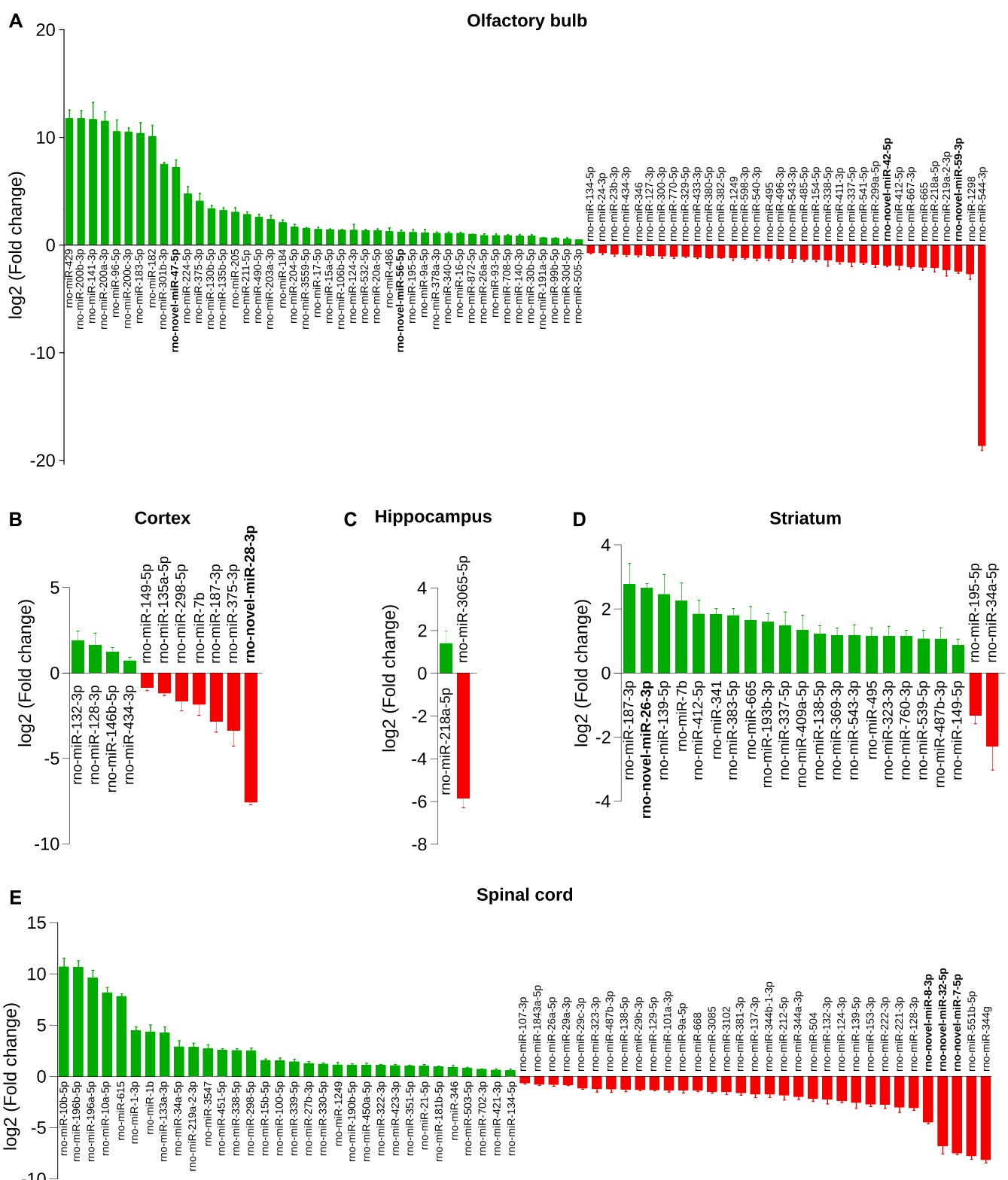

**Figure 5.   Differentially expressed miRNAs between the structures.**
All the depicted miRNAs are statistically dysregulated in the structure considered compared with all other structures as assessed with the DESeq2 algorithm and Wald test (P < 0.05). The log2 of the fold change of over- (green) and down-regulated (red) miRNAs is indicated; miRNAs written in bold correspond to novel miRNAs.
**(A)** The olfactory bulb shows the highest number of differentially regulated miRNAs: 44 are up-regulated and 34 are down-regulated. In the cortex **(B)**, four miRNAs are significantly up-regulated and seven are down-regulated, whereas in the hippocampus **(C)**, only one miRNA is overexpressed and one is underexpressed. **(D)** In the striatum, 20 and 2 miRNAs are up- and down-regulated, respectively. **(E)** In the spinal cord, 32 and 34 miRNAs are up- and down-regulated, respectively.

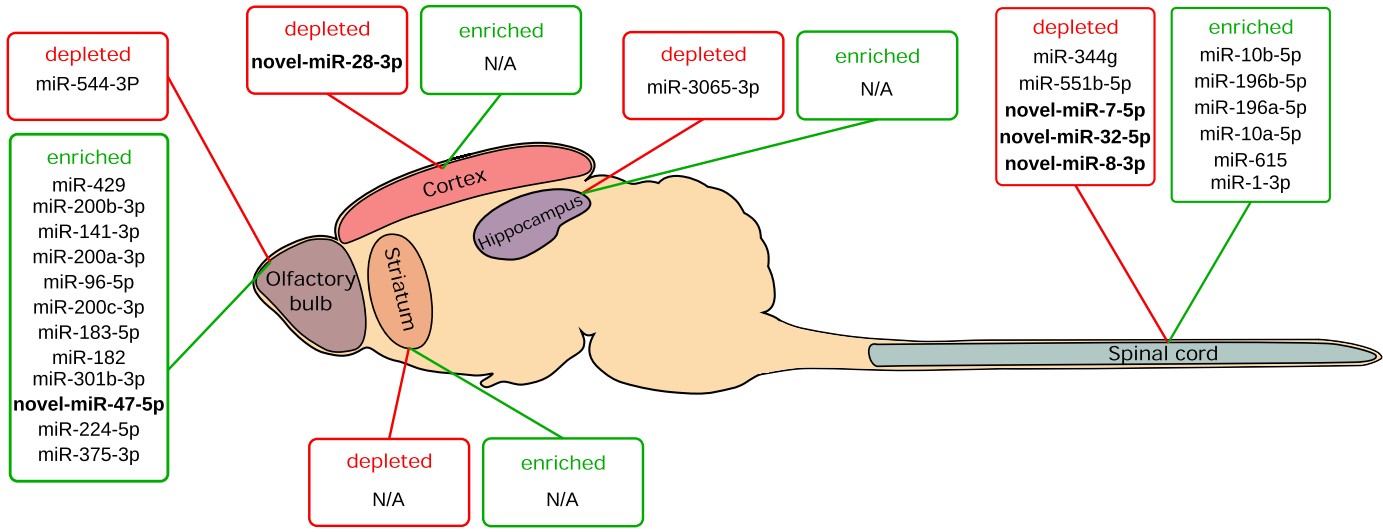

**Figure 6. Enrichment/depletion of miRNAs in specific structures.**
To focus on the miRNAs that could have the most relevant role in transcriptome regulation, we defined as enriched miRNAs those with a fold change >16, and depleted miRNAs those with a fold change <−16. Except for the striatum, all structures express specifically enriched or depleted miRNAs. The olfactory bulb shows the highest number of specific miRNAs (13 miRNAs), the spinal cord expresses 11 specific miRNAs, the cortex and the hippocampus exhibit only one specifically depleted miRNA, and interestingly, the cortex-specific miRNA is a novel miRNA (novel miRNAs are written in bold).

up-regulated were shown (Fig 7D). For instance, the expression of Cdhr1 (log2FC = 6.02), a cadherin-related protein, the transcription factors Sp8 (log2FC = 4.97) and Barhl2 (log2FC = 3.79), the neurogenesis marker Doublecortin (Dcx, log2FC = 3.55), and Wnt5a (log2FC = 3.39) were highly up-regulated in the olfactory bulb compared with all other structures studied. GO term enrichment suggests that the regulated miRNA-mRNA pairs could have a role in the specification of this brain structure (Fig 7E). Indeed terms such as "axon extension", "regulation of anatomical structure size," and "chemical synaptic transmission" were statistically enriched.

In the cortex, only one miRNA, the newly discovered novel-miR-28-3p, was specifically depleted (log2FC = −7.55), and the cumulative frequency distribution of expression changes of all its predicted targets was significantly shifted to the right compared with non-target genes, revealing a global up-regulation (Fig 8A). Twenty of novel-miR-28-3p predicted targets were statistically up-regulated (Fig 8B), for example, Kcns1 (log2FC = 5.27), a voltage-gated potassium channel subunit, Vip (log2FC = 2), a neurotransmitter and neuromodulator with neurotrophic properties, cerebellin 2 (Cbln2, log2FC = 1.96), a protein forming homohexamers at the presynaptic compartment, and Lynx1 (log2FC = 1.2), a protein associated with the cholinergic regulation. GO term analysis revealed neuroscience-familiar keywords such as "potassium ion transport" and "actin cytoskeleton organization" (Fig 8C).

In the hippocampus, the specific depletion of miR-3065-5p (log2FC = −5.85) was associated with the up-regulation of many of its predicted targets. Hence, the cumulative frequency distribution of expression changes of all its predicted targets was significantly shifted to the right compared with nontarget genes, revealing an overall up-regulation (Fig 9A). Statistical analysis showed that 70 target genes of miR-3065-5p were significantly up-regulated in the hippocampus compared with all other CNS structures (Table S10). Among these, many have a known role in

neuronal function such as, the LIM homeobox 9 protein (Lhx9, log2FC = 4.53), the interleukin IL-16 (log2FC = 3.6), the gonadotrophin-releasing hormone receptor (Gnrhr, log2FC = 3.55), the protein kinase C γ (Prkcg, log2FC = 2.5), the protocadherin-19 (log2FC = 1.88), and the GluA1 AMPA receptor (Gria1, log2FC = 1.85), Fig 9B. In addition, these enriched mRNAs were associated with relevant GO terms such as "negative regulation of nervous system development" (Fig 9C).

Finally, concerning specifically enriched miRNAs in the spinal cord, the cumulative frequency distribution of expression changes of all predicted targets was significantly different from nontarget genes (Fig 10A). Although most of the predicted targets showed an expected down-regulation, some were up-regulated. Among the predicted targets, 71 were statistically down-regulated (Table S10). Fig 10B shows the 20 most down-regulated mRNAs, such as the transcription factor Bcl11b (also known as Ctip2, log2FC = −4.08) targeted by both miR-10b-5p and miR-615, the vesicular glutamate transporter 1 (Slc17a7, log2FC = −5.44), the transcription factor subunit FosB (log2FC = −4.67), the activity-regulated cytoskeleton-associated protein (Arc, log2FC = −4.25), or the synaptopodin protein (Synpo, log2FC = −3.83). Concerning specifically depleted miRNAs in the spinal cord, the cumulative frequency distribution of expression changes of all predicted targets was significantly different from nontarget genes (Fig 10C). Although most of the predicted targets showed an expected up-regulation, some were down-regulated. Among the predicted targets, 91 were statistically up-regulated (Table S10). Hence, many predicted targets of miR-344 g (log2FC = −8.11) were up-regulated (Fig 10D) such as the transcription factors paired box gene 2 (Pax2, log2FC = 7.0), Lmx1b (log2FC = 4.69), and Lbx1 (log2FC = 4.62). The glycine neurotransmitter transporter Slc6a5 (log2FC = 6.51) and the apelin receptor (Aplnr, log2FC = 4.35) were also up-regulated. miR-551b-5p was another miRNAs strongly depleted (log2FC = −7.74) in the spinal cord, and its predicted targets

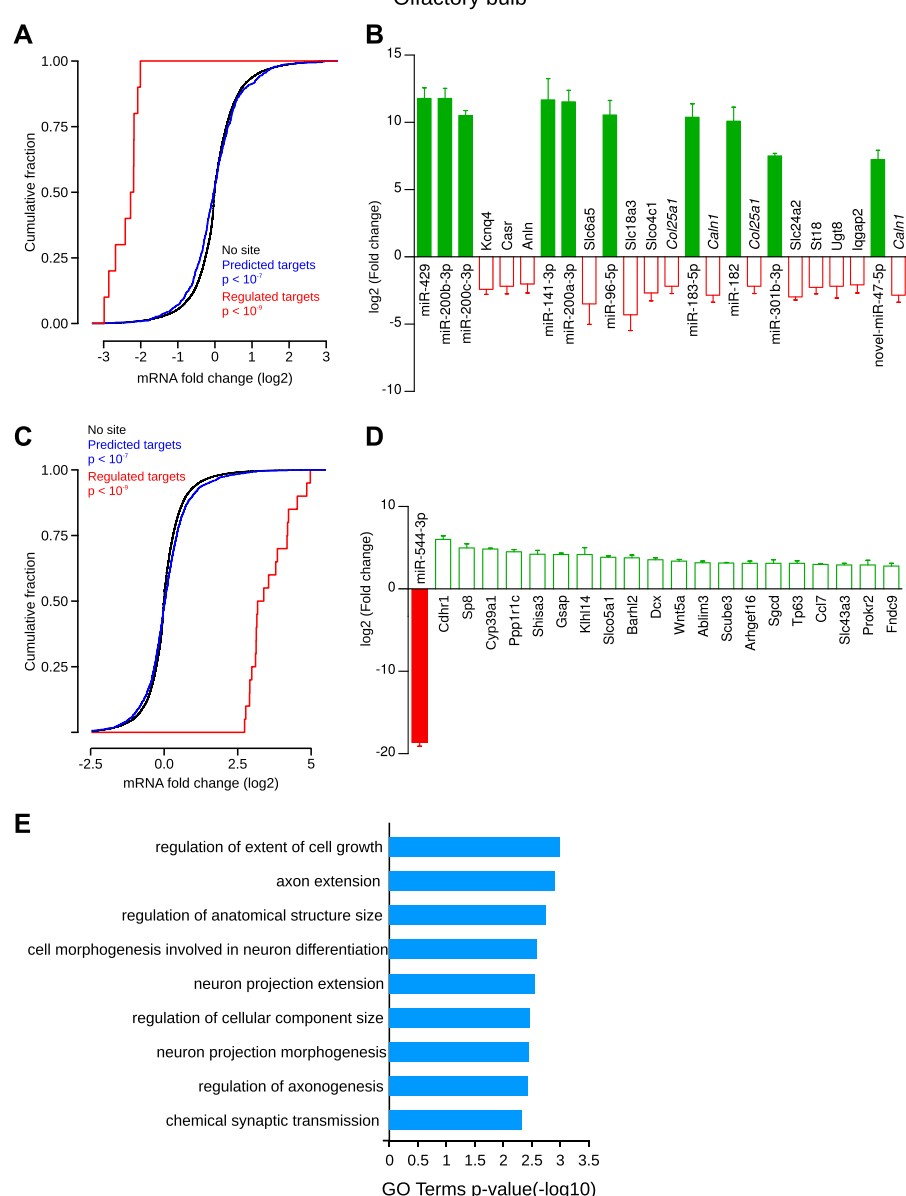

Figure 7.  Olfactory bulb–specific miRNAs and their regulated targets.
(A) Concerning specifically enriched miRNAs, cumulative frequency distribution of expression changes of all predicted targets is significantly different from nontarget genes (blue line versus black line, Kolmogorov–Smirnov test). Distribution of expression changes of statistically regulated target genes (red) shows a down-regulation compared with nontarget genes. (B) Individual gene statistical analysis revealed that 12 target genes of the enriched miRNAs are statistically down-regulated (DESeq2 algorithm and Wald test, P < 0.05). Interestingly, some regulated genes are predicted to be the target of multiple miRNAs (gene names written in italics). Data shown are mean ± SD form three biological replicates. (C) Concerning specifically depleted miRNAs, cumulative frequency distribution of expression changes of all predicted targets is significantly different from nontarget genes (blue line versus black line, Kolmogorov–Smirnov test). Distribution of expression changes of statistically regulated target genes (red) shows an up-regulation compared with nontarget genes. (D) Individual gene statistical analysis revealed that 40 target genes of miR-544-3p are up-regulated (DESeq2 algorithm and Wald test, P < 0.05); only the 20 most regulated are depicted. Data shown are mean ± SD form three biological replicates. (E) GO term enrichment analysis of the selected targets.

were largely up-regulated, for instance, the transcription factors Hoxb3 (log2FC = 6.91), Pax8 (log2FC = 6.67), Nkx6-1 (log2FC = 4.65), and Lhx4 (log2FC = 4.31), polycystic kidney disease 2–like 1 protein (Pkd2l1 also known as Trpp3, log2FC = 5.51), a cationic channel, and the glucagon-like peptide 1 receptor (Glp1r, log2FC = 5.06). GO term analysis highlighted many keywords relevant in the neuroscience and neurodevelopment field such as "pattern specification process", "spinal cord development" and "chemical synaptic transmission" (Fig 10E).

# Discussion

The aim of this study was to determine the miRNome of five major structures of the CNS, the cortex, the hippocampus, the olfactory bulb, the striatum, and the spinal cord, and to evaluate the impact of enriched/depleted miRNAs on the transcriptome.

Three principal methods are used to measure the expression levels of miRNAs in a tissue: reverse transcription quantitative PCR (RT-qPCR) (Chen et al, 2005; Shi & Chiang, 2005), microarray hybridization (Yin et al, 2008), and small RNA-Seq (Hafner et al, 2008), all of which face unique challenges compared with their use in miRNA profiling. However, RNA-Seq offers competitive advantages, such as enabling discovery of new sequences variants and new sequences (Willenbrock et al, 2009). Consequently, we performed small RNA-Seq to obtain an exhaustive analysis of the miRNome of these five CNS structures, and we found a total of 90 novel miRNAs.

Previous studies used small RNA-Seq to analyze the miRNome of the rat brain. Unfortunately, all these studies analyzed miRNA expression in only one brain structure, occluding any attempt to

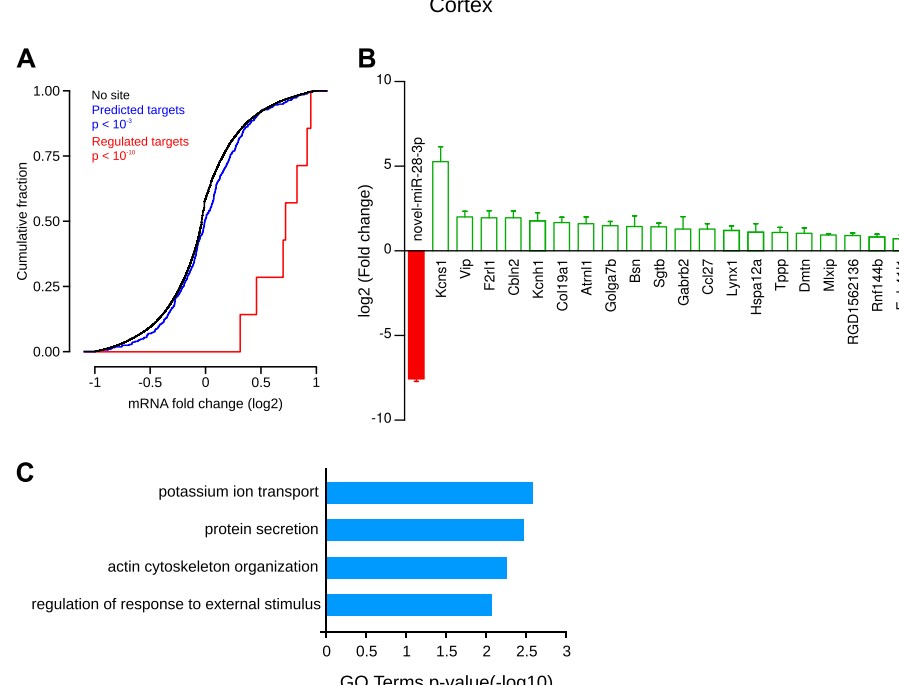

Cortex

**Figure 8. Cortex-specific miRNA and its regulated targets.**
**(A)** Cumulative frequency distribution of expression changes of all predicted targets is significantly shifted to the right compared with nontarget genes, revealing a global up-regulation (blue line versus black line, Kolmogorov–Smirnov test). Distribution of expression changes of statistically regulated target genes (red) shows an up-regulation compared with nontarget genes. **(B)** Individual gene statistical analysis revealed that 20 target genes of novel-miR-28-3p are up-regulated (DESeq2 algorithm and Wald test, *P* < 0.05). Data shown are mean ± SD form three biological replicates. **(C)** GO term enrichment analysis of the selected targets.

identify structure-specific miRNAs. For instance, Shinohara et al (2011) focused their attention on the hippocampus and overall found a relatively modest number of novel miRNAs (17), which was restricted to two candidates when they applied a filter based on the abundance. None of the two miRNA candidates identified by Shinohara et al (2011) were found in our data. In addition, Amar et al (2012) analyzed the expression of miRNAs in the hypothalamus only, and the number of reads they obtained for each sample was not compatible with novel miRNA discovery. Yao et al analyzed the miRNome of the developing cerebral cortex from E10 to P28 (Yao et al, 2012) found 101 potential novel miRNAs. Finally, Yin et al used as starting material the whole brain with all structures mixed together and found 171 candidate novel miRNAs. Unfortunately, the novel miRNAs discovered in Yin's study were not yet implemented in miRBase. However, we aligned the sequences of their novel miRNAs against ours and found matches for five miRNAs. Thus, Yin's novel-miR-16, novel-miR-18, novel-miR-20, novel-miR-32, and novel-miR-33 match with ours novel-miR-79-3p, novel-miR-64-5p, novel-miR-38-5p, novel-miR-87-5p, and novel-miR-67-5p, respectively.

All the computational tools developed for identifying miRNAs from small RNA-Seq data sets suffer from high false-positive and false-negative rates and also of a lack of consistency across species (Li et al, 2012; Williamson et al, 2013; Kang & Friedländer, 2015). A strategy to limit the number of false-positive novel miRNAs and to focus on the most biologically relevant ones could be to apply a filter on the predicted novel miRNAs based on the abundance. Thus, we initially obtained more than 8000 novel miRNAs from our samples with the miRPro algorithm (based on miRDeep2), and we reduced this number to 90 by applying two filters: the number of reads (>1 cpm of total reads) and the presence of the novel miRNA

in at least two independent samples. The number of novel miRNAs discovered by small RNA-Seq experiments is hard to predict because it depends on many parameters such as the diversity of the nature of samples (culture, tissues, and fluids), the number of reads for each sample, the quality of the library, and finally the bioinformatic tools to predict novel miRNA and the efficiency to discard false-positive novel miRNAs. In this regard, our results are in accordance with the literature. Indeed, using miRDeep2 as miRNA-prediction algorithm, recent studies found from 71 novel miRNAs in the mouse brain after an infection with Japanese encephalitis viruses (Li et al, 2016), up to 171 novel miRNAs for the Yin's study (Yin et al, 2015).

In comparison with the miRNA database (miRBase) of the mouse (*Mus musculus*, 1,915 mature miRNAs and 1,193 precursors) and human (*Homo sapiens*, 2,558 mature miRNAs and 1,881 precursors), the content of the rat miRNA database is low (*Rattus norvegicus*, 765 mature miRNAs and 495 precursors). This may reflect the fact that the human and the mouse are two species intensively studied from a genetic point of view compared with the rat. Even if we implement our novel miRNAs in miRBase, the rat miRBase would still be more than half less referenced than the mouse one and three times less referenced than the human one. This may be due to the fact that we focused our study only on the CNS and that we may have discovered more miRNAs if we had made a pan-organism study. Nevertheless, our work significantly increases the knowledge on rat brain miRNA expression.

We searched orthologous sequences for our novel miRNAs in the mouse and human genomes because miRNAs that are well conserved across species often play crucial roles. Among the 90 novel miRNAs discovered, 12 have orthologous sequences in mouse and/or human organisms (we determined orthologs by a minimum of

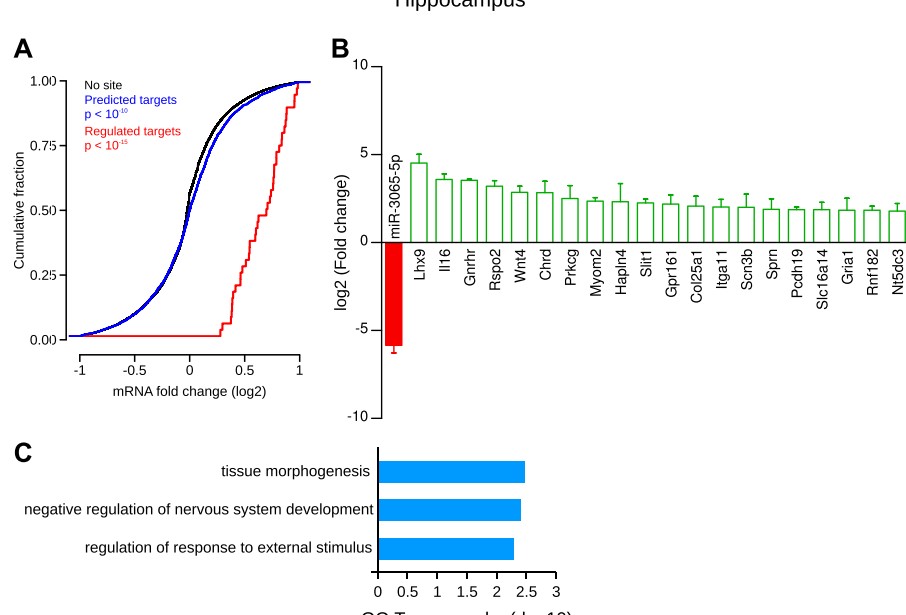

**Figure 9. Hippocampus-specific miRNA and its regulated targets.**
**(A)** Cumulative frequency distribution of expression changes of all predicted targets is significantly shifted to the right compared with nontarget genes, revealing a global up-regulation (blue line versus black line, Kolmogorov–Smirnov test). Distribution of expression changes of statistically regulated target genes (red) shows an up-regulation compared with nontarget genes. **(B)** Individual gene statistical analysis revealed that 70 target genes of novel-miR-28-3p are up-regulated (DESeq2 algorithm and Wald test, $P < 0.05$); only the 20 most regulated are depicted. Data shown are mean ± SD form three biological replicates. **(C)** GO term enrichment analysis of the selected targets.

95% of similarities between the sequences). Thus, most of these newly discovered miRNAs in the rat brain have no homologue in the mouse or human. Because mouse and human brains had already been analyzed for new miRNAs, it may suggest that these miRNAs are rat specific, supporting rat-specific gene regulations. Studies on human genome suggest that recently discovered miRNAs are mainly evolutionarily young (Friedländer et al, 2014). In addition, it has been recently shown that within a species, interstrain variation exists with functional significance on the targeted mRNAs (Trontti et al, 2018). However, we cannot exclude that the mouse and human orthologs of these miRNAs have not been found yet due to the bioinformatic tools used for miRNA prediction. For example, both novel-miR-7-5p and novel-miR-8-3p are orthologs of the mouse miR-344-5p and miR-344h-3p, respectively. Interestingly, a work showed by RT–qPCR and in situ hybridization that miR-344-3p is detected in cortical regions surrounding the ventricular system, choroid plexus, glomerular layer of the olfactory bulb, and granular cell layer of the cerebellar cortex (Liu et al, 2014). They showed that miR-344-3p is neural specific during mouse embryonic development and that it may play an important role in CNS morphogenesis. In our analysis, we found that novel-miR-7-5p and novel-miR-8-3p are specifically down-regulated in the spinal cord, but expressed in all the other structures, approximately at the same level. It should be interesting to define more specifically the spatial and temporal distribution of these novel-miRNAs to compare with mouse miR-344 expression and function.

Furthermore, among the 90 novel miRNAs discovered, we found 23 miRNAs that we called "functional orthologs", meaning that they share the same seed region with other miRNAs. Because miRNA target prediction is based on the complementarity between the seed region of the miRNA and the 3'UTR of the target mRNA, we hypothesized that these "functional orthologs" would have the same targets. Indeed, miRNAs sharing the same seed region often

share the same targets and are grouped as a family of miRNAs (Ambros et al, 2003). To test this hypothesis, we designed a proof-of-concept experiment based on a luciferase assay using the novel-rno-miR-21-5p that was predicted to be a "functional ortholog" of mouse mmu-miR-676-3p. We used LCE2D, a predicted target gene of mmu-miR-676-3p sharing with novel-rno-miR-21-5p the same interaction based on the seed region only. Luciferase experiment showed that both mmu-miR-676-3p and novel-rno-miR-21-5p were able to inhibit the expression of the luciferase-LCE2D construct. This result can be considered a clue that "functional orthologs" may regulate the same targets. However, it cannot be assumed from one experiment that they are functionally equivalent and further reporter gene assays have to be performed to confirm the targets of all "functional orthologs".

The increasing amount of small RNA-Seq data provides a more complete view of the miRNome, with the discovery of novel canonical miRNAs, and many other miRNA-like small RNAs that originate from non-canonical miRNA genes. In addition, the increasing number of novel miRNAs discovered across the literature, sometimes repeatedly such as Yin's work and ours, increases the complexity of miRNA indexation. These arguments enforce to revise miRNA's nomenclature conventions to ease the comparison of miRNA research across species (Desvignes et al, 2015). Characterization of 5'-end sequence of miRNAs is important because this so-called seed region of miRNAs is critical in miRNA-target mRNA binding (Engels & Hutvagner, 2006). Analysis of nucleotide sequences in eukaryotic miRNAs showed a clear bias for U or A at the 5' position (Frank et al, 2010). Here, we found that U was the most representative nucleotide in the 5' position of our novel miRNA sequences, followed by C and A. Results from others studies on other organisms are consistent (Zhou et al, 2009; Wei et al, 2011; Ji et al, 2012; Pacholewska et al, 2016). This result reinforces the biological pertinence of our novel miRNAs.

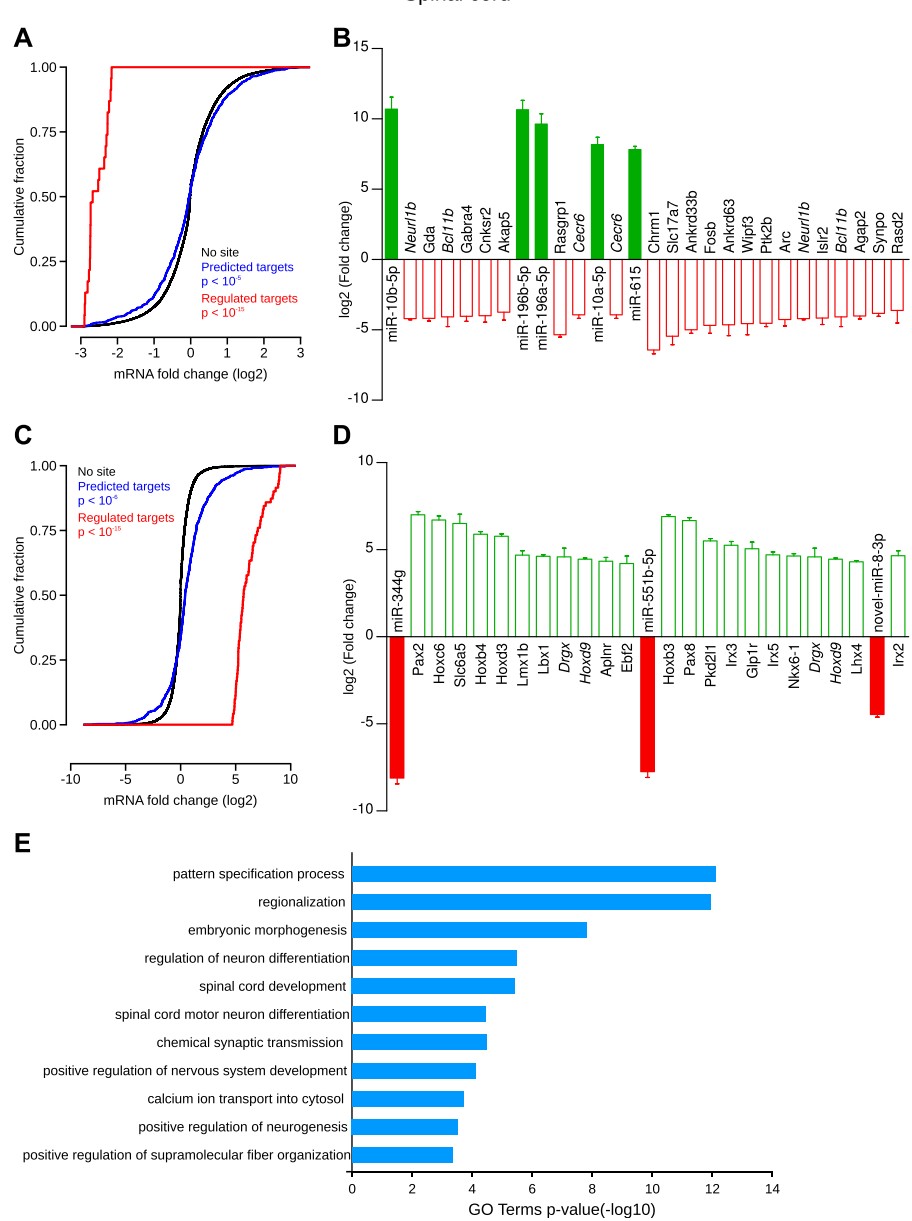

**Figure 10.  Spinal cord–specific miRNAs and their regulated targets.**
**(A)** Concerning specifically enriched miRNAs, cumulative frequency distribution of expression changes of all predicted targets is significantly different from nontarget genes (blue line versus black line, Kolmogorov–Smirnov test). Distribution of expression changes of statistically regulated target genes (red) shows a down-regulation compared with nontarget genes. **(B)** Individual gene statistical analysis revealed that 71 target genes of the enriched miRNAs are statistically down-regulated (DESeq2 algorithm and Wald test, $P < 0.05$); only the 20 most regulated are depicted. Interestingly, some regulated genes are predicted to be the target of multiple miRNAs (gene names written in italics). Data shown are mean ± SD form three biological replicates. **(C)** Concerning specifically depleted miRNAs, cumulative frequency distribution of expression changes of all predicted targets is significantly different from nontarget genes (blue line versus black line, Kolmogorov–Smirnov test). Distribution of expression changes of statistically regulated target genes (red) shows an up-regulation compared with nontarget genes. **(D)** Individual gene statistical analysis revealed that 91 target genes of the depleted miRNAs are up-regulated (DESeq2 algorithm and Wald test, $P < 0.05$); only the 20 most regulated are depicted. Interestingly, some regulated genes are predicted to be the target of multiple miRNAs (gene names written in italics). Data shown are mean ± SD form three biological replicates. **(E)** GO term enrichment analysis of the selected targets.

It is well described that tissue- and cell-specific expression patterns of miRNAs are responsible for their particular biological function within a specific system. For example, the most famous brain-enriched miRNA, miR-124, defines brain-specific gene expression in human cells (Lim et al, 2005).

The rationale of our study was to find structure-specific miRNAs that could regulate a specific set of genes in a given CNS structure. We reasoned that to have a strong impact on gene expression and finally on the physiological function of the different CNS structures, specific miRNAs should be highly regulated in one structure compared with all other ones. Thus, we selected as specifically enriched miRNAs, those with a log 2 of the fold-change greater than 4, meaning that the expression of the given miRNA was at least 16

times higher in this structure than in any other structure. In addition to specifically enriched miRNAs, the specific depletion of miRNAs in a structure could have a strong impact on CNS physiology by unleashing gene expression of specific targets. Therefore, we selected as depleted miRNAs, miRNAs which had a log 2 of the fold change lower than −4, corresponding to, at least, a 16 times decrease in miRNA expression. This analysis revealed specific miRNAs for each structure except for the striatum. To evaluate the potential impact of these specific miRNAs, we measured the expression of their predicted target mRNAs. Thus, we quantified the expression of all mRNAs using mRNA-Seq on the very same samples used for miRNA quantification. After statistical analysis of the differentially regulated mRNAs, we correlated miRNA and mRNA expression data

using target predictions as determined by TargetScan for known miRNAs and miRDB tools for novel miRNAs. To evaluate whether the predicted target genes were more regulated than nontarget genes, we analyzed the cumulative frequency distribution of expression change of all miRNA target genes versus nontarget genes. This analysis showed that the expression of predicted targets is significantly different from the one of nontargeted genes. Although in most cases the expression of the predicted targets was regulated as expected (opposite to miRNA expression because miRNAs act as inhibitors), on some occasion the predicted targets showed an unexpected regulation. This result is not surprising because miRNAs are not the only regulators of gene expression. Hence, other mechanisms, such as transcription factors, affect mRNA levels, and the net effect on mRNA expression is a combination of all these mechanisms. To try to focus on the most biologically relevant targets for each structure, we selected the mRNAs displaying the most important regulation (either the 20 uppest or 20 lowest regulated mRNAs). GO enrichment analysis of these top 20 mRNAs revealed that most of the regulated targets were genes known for their involvement in the development or the function of the nervous system. In addition, literature review showed that some of these targets specifically regulated in a CNS structure were already known to have a specific role in the given structure. Table 4 summarizes the correlation between our results and the literature.

We found 12 enriched miRNAs in the olfactory bulb, and among them the expression of all the members of the well-described miR-200 family (Senfter et al, 2016) composed of miR-200a, b, c, and miR-429. Recent studies demonstrate that the miR-200 family has an important role for the regulation of the proliferation and the differentiation of neuronal cells (Pandey et al, 2015; Beclin et al, 2016). More interestingly, a work incriminates the miR-200 family in the regulation of the olfactory neurogenesis (Choi et al, 2008), which is in accordance with our result showing an over-representation of all the members of this family of miRNAs in the olfactory bulb. In addition, we found that miR-544-3p was strongly down-regulated and among its predicted targets several genes are highly up-regulated. For instance, Sp8, a zinc-finger transcription factor known to regulate olfactory bulb interneuron development (Li et al, 2018) Bar H Like Homeobox 2 (Barhl2), another transcription factor, is known to play a key role in the development of region-specific differences in embryonic mouse forebrain through interaction with Pax6 (Parish et al, 2016) and with the diencephalon patterning by regulating Shh (Ding et al, 2017). In addition, Doublecortin (Dcx) is mandatory for proper establishment of the olfactory tract (Belvindrah et al, 2011), and Wnt5a is required for the development of the olfactory bulb (Zaghetto et al, 2007; Paina et al, 2011; Pino et al, 2011).

In the cortex, the novel-miR-28-3p is specifically depleted, and cortex-relevant predicted targets are highly up-regulated such as the vasoactive intestinal peptide (Vip). Vip-expressing interneurons have a key role in cortical circuit development, suggesting a possible contribution to pathophysiology in neurodevelopmental disorders (Batista-Brito et al, 2017). Cerebellins are secreted hexameric proteins that form tripartite complexes with the presynaptic cell-adhesion molecules neurexins and the postsynaptic glutamate receptor–related proteins GluD1 and GluD2. These tripartite complexes are believed to regulate synapses. In particular, cerebellin 2

(Cbln2) is specifically expressed in a subpopulation of excitatory cortical neurons (Seigneur & Südhof, 2017). Finally, Lynx1 is a cholinergic brake involved in the maintenance of the stability of cortical networks (Morishita et al, 2010).

In the hippocampus, miR-3065-5p was strongly down-regulated, and among its putative targets, many were up-regulated and related to hippocampal functions. Hence, the developmental regulatory gene Lhx9 seems to be involved in the development of the hippocampal subdivisions (Abellán et al, 2014). The neuronal interleukin-16 (Il16), a PDZ-containing protein, enriched in the hippocampus is responsible for the modulation of Kv4.2K+ currents, thereby regulating the intrinsic neuronal properties (Fenster et al, 2007). In addition, the gonadotrophin-releasing hormone receptor (Gnrhr) was up-regulated, and a recent study strongly suggests a role of specific Gnrhr activation in neuronal plasticity in the hippocampus (Schang et al, 2011). Plasticity mechanisms such as long-term potentiation are also supported by a regulated target of miR-3065-5p, namely the proteinase C γ (Prkcg) (Gärtner et al, 2006). Non-clustered protocadherins (PCDHs) are calcium-dependent adhesion molecules hypothesized to be involved in neuronal circuit formation and plasticity. Thus, the protocadherin-19 (Pcdh19), a potential target of miR-3065-5p, is regulated in the hippocampus in response to a plasticity paradigm (Kim et al, 2010), and mutations on Pcdh19 are responsible for epilepsy and mental retardation (Dibbens et al, 2008). Finally, the GluA1 AMPA receptor, a key element of homeostatic plasticity in the hippocampus (Sutton et al, 2006), was up-regulated. Interestingly, we previously showed that GluA1 fine-tuning involved miRNA regulations (Letellier et al, 2014).

Finally, in the spinal cord, we found both specifically enriched and depleted miRNAs. We found six miRNAs specifically enriched in the spinal cord. Two families are represented: miR-196 family, represented by miR-196a and b (Chen et al, 2011), and miR-10 family, represented by miR-10a and b (Tehler et al, 2011). Interestingly, some studies demonstrate the role of the miR-196 in the regulation of the motor neuron programs (Asli & Kessel, 2010). Furthermore, our results are in accordance with the literature: miR-10a and miR-10b that are specifically expressed in the spinal cord in our study are also found specific of this region in the mouse (Bak et al, 2008). One of the predicted targets of miR10b-5p was down-regulated and known for its function in the spinal cord. Hence, in loss-of-function experiments, it has been shown that Bcl11b plays a critical role in the development of axonal projections to the spinal cord in vivo (Arlotta et al, 2005). miR-615 was another enriched miRNA, and interestingly, among its regulated targets, many were associated with the pain-processing mechanisms in the spinal cord. Hence, expression of the vesicular glutamate transporter 1 (Vglut1 also known as Slc17a7) defines a subpopulation of neurons involves in pain processing (Landry et al, 2004) and nerve injury induces a decrease in Vglut1 expression in the dorsal horn of the spinal cord (Brumovsky et al, 2007). The expression of the fos-related fosB gene, sharing many properties with c-Fos, is increased during spinal cord development (Redemann-Fibi et al, 1991), and interestingly, fosB is involved in the transcriptomic cascade triggered during nociception (Herdegen et al, 1991). In addition, the expression of the Arc/Arg3.1 protein is specifically enhanced in response to nociceptive stimulations (Hossaini et al, 2010) through the activation of the intracellular mGluR5 receptor (Vincent et al, 2016). Finally, we

**Table 4.** Relevance of the correlation between miRNA and mRNA expression.

| Structure | Specific miRNA | Regulated target mRNA | mRNA function associated with the structure |
|---|---|---|---|
| Olfactory bulb | *miR-544-3p* | **Sp8** | Transcription factor known to regulate olfactory bulb interneuron development (Li et al, 2018) |
| | | **Barhl2** | Transcription factor involved in the development of region-specific differences in the forebrain (Parish et al, 2016) and the diencephalon (Ding et al, 2017) |
| | | **Dcx** | Mandatory for proper migration and development of olfactory bulb neurons (Belvindrah et al, 2011) |
| | | **Wnt5a** | Necessary for olfactory axon connections (Zaghetto et al, 2007; Paina et al, 2011; Pino et al, 2011) |
| Cortex | *novel-miR-28-3p* | **Vip** | Vip-expressing interneurons are crucial for cortical circuits development (Batista-Brito et al, 2017) |
| | | **Cbln2** | Expressed in the subpopulation of excitatory cortical neurons (Seigneur & Südhof, 2017) |
| | | **Lynx1** | Mandatory for cortical network stability (Morishita et al, 2010) |
| Hippocampus | *miR-3065-5p* | **Lhx9** | Implicated in the development of the hippocampal subdivisions (Abellán et al, 2014) |
| | | **Il16** | Modulation of Kv4.2K+ currents and thus neuronal intrinsic properties (Fenster et al, 2007) |
| | | **Gnrhr** | Involved in the hippocampus-specific neuronal plasticity mechanism (Schang et al, 2011) |
| | | **Prkcg** | Supports hippocampal long-term potentiation (Gärtner et al, 2006) |
| | | **Pcdh19** | Regulated in plasticity paradigm (Kim et al, 2010) and mutations linked to epilepsy and mental retardation (Dibbens et al, 2008) |
| | | **Gria1** | Involved in neuronal homeostatic plasticity (Sutton et al, 2006; Letellier et al, 2014) |
| Spinal Cord | **miR-10b-5p** | *Bcl11b* | Mandatory for the development of axonal projections (Arlotta et al, 2005) |
| | **miR-615** | *Slc17a7* | Defines a subpopulation of neurons involved in pain processing (Landry et al, 2004; Brumovsky et al, 2007) |
| | | *FosB* | Increased expression during spinal cord development and nociception (Herdegen et al, 1991; Redemann-Fibi et al, 1991) |
| | | *Arc* | Enhanced expression in response to pain (Hossaini et al, 2010) |
| | | *Synpo* | Regulated in bone cancer pain conditions (Elramah et al, 2017) |
| | *miR-344g* | **Pax2** | Involved in spinal cord development (Larsson, 2017) |
| | | **Lmx1b** | Involved in spinal cord development (Ding et al, 2004; Hilinski et al, 2016) |
| | | **Lbx1** | Involved in pain mechanisms (Gross et al, 2002; Cheng et al, 2017) |
| | | **Aplnr** | Involved in pain mechanisms (Xiong et al, 2017) |
| | *miR-551b-5p* | **Hoxb3** | Involved in spinal cord development (Yau et al, 2002) |
| | | **Pax8** | Involved in spinal cord development (Batista & Lewis, 2008) |
| | | **Nkx6-1** | Involved in spinal cord development (Sander et al, 2000; Vallstedt et al, 2001) |
| | | **Glp1r** | Involved in pain mechanisms (Djenoune et al, 2014) |
| | | **Pkd2l1** | Defines a subpopulation of neurons regulating locomotion (Böhm et al, 2016) |

For each structure, specifically enriched or depleted miRNAs (in bold or italic, respectively) were associated with oppositely regulated mRNAs (up-regulated in bold and down-regulated in italic) known for their role in the given CNS structure.

previously demonstrated that synaptopodin (Synpo), a key element of the spine apparatus, was regulated in the spinal cord in an animal model of bone cancer pain (Elramah et al, 2017).

Concerning depleted miRNAs, down-regulation of miR-344 g and miR-551b-5p was associated with a strong up-regulation of many of their predicted targets involved in the development of the spinal cord; for instance, Pax gene members, Pax2 and Pax8 (Batista & Lewis, 2008; Larsson, 2017), Hoxb3 (Yau et al, 2002), the Nkx homeobox gene member Nkx6-1 (Sander et al, 2000; Vallstedt et al, 2001), and the LIM homeobox transcription factor 1–β (Lmx1b [Ding et al, 2004; Hilinski et al, 2016]). Again, in the regulated targets, several were involved in pain mechanisms such as the homeodomain factor Lbx1 (Gross et al, 2002; Cheng et al, 2017), the apelin receptor (Aplnr [Xiong et al, 2017]), or the glucagon-like peptide-1 receptor (Glp1r [Gong et al, 2014a, 2014b]). Finally, a predicted target of miR-551b-5p, the transient receptor potential channel Pkd2l1, was up-regulated. Interestingly, Pkd2l1 is a specific marker of spinal cerebrospinal fluid–contacting neurons (Djenoune et al, 2014), an evolutionarily conserved population of neurons regulating locomotion by relaying mechanical stimuli to spinal circuits (Böhm et al, 2016).

Although the link between the enriched/depleted miRNAs and the regulated mRNAs in this study is only correlative and based on target predictions, the GO term analysis and the relevance of the literature associated with these targets strongly suggest that the specific miRNAs may play a role in the transcriptomic specificity of the CNS structures. The miRNAs playing the most important role on CNS structure function might be the specifically depleted miRNAs. Thus, by unleashing gene expression of their target mRNAs, they may play an important role in the specification and the function of the specific structures of the rat CNS. In the future, it would be interesting to artificially modify the expression of these miRNAs during development to confirm their role in the specification of the structures of the CNS. In that perspective, our work will help further studies in dissecting biological functions of miRNAs in the CNS.

# Materials and Methods

### Animals

Three male Wistar rats aged 12 wk were dissected to collect CNS tissues. All experimental procedures followed the ethical guidelines of the University of Bordeaux's ethics committee. Animals were anesthetized with an intraperitoneal injection of pentobarbital and then perfused with aCSF (NaCl 130.5 mM; KCl 2.4 mM; MgSO$_4$ 1.3 mM; KH$_2$PO$_4$ 1.2 mM; Hepes 1.25 mM; CaCl$_2$ 2.4 mM; Glucose 10 mM; NaHCO$_3$ 19.5 mM) to wash out blood from the nervous tissue and avoid contaminations with blood cells RNA.

### Sample collection and RNA extraction

From each rat, five structures of the CNS (the olfactory bulb, a part of the cortex, the hippocampus, the striatum, and the dorsal spinal cord) were dissected and immediately placed into a 1.5-ml tube containing 700 µl of TRIzol reagent (QIAGEN) on ice. To avoid cross-contamination, we used new disposable dissecting instruments for each sample. Samples were crushed with the Fast Prep 24

instrument (MP Biomedicals). Total RNA was extracted following the manufacturer's instructions, allowing isolation of all RNAs including the small RNAs (miRNeasy Micro Kit; QIAGEN). The quality and the quantity of the RNA samples were determined with a spectrophotometer NanoDrop One (Ozyme) and with an RNA chip (Agilent RNA 6000 Nano kit) into the BioAnalyzer 2100 (Agilent).

### Preparation and sequencing of small RNA libraries

The libraries were synthesized from 1 µg of RNA for each sample, following the manufacturer's instructions (NEXTflex Illumina Small RNA Seq Prep; Bioo Scientific Corporation). Briefly, 3′ adenylated adaptaters, followed by 5′ adaptaters, were sequentially added to RNA strands. Then, RNA samples were reverse transcribed using the 3′ adaptaters as the template for the RT primer. Finally, cDNAs were amplified by PCR. Then, the quality and the quantity of cDNA were analyzed with a DNA chip (Agilent).

A size selection of the cDNA libraries (150 pb) was performed with a Pippin Prep (Sage Science). All the samples were pooled in an equimolar manner and purified on column (MinElute PCR Purification Kit; QIAGEN). Finally, the quality and quantity of cDNA were checked with a DNA chip (Agilent) before sequencing. The samples were sent to the functional genomic platform of Nice Sophia Antipolis, France (https://www.france-genomique.org). The next-generation single-read sequencing was performed using the sequencer NextSeq 500, Illumina, using the single-end 75-bp high output sequencing mode. We selected this mode to achieve a minimum of 8 millions of reads per sample to enable accurate detection of novel miRNAs with the miRPro algorithm. The preliminary results are the following:

Total reads: 454,577,868.
Filtered too short reads (size < 15 b): 1,150,734 (0.3% of total).
Not filtered reads:

1) reads ready to be mapped with Bowtie2: 453,427,134 (99.7% of total)
2) unmapped reads, with low quality mm > 2 or contamination reads: 17,632,800 (3.9% of unfiltered reads)
3) mapped reads

   -ready to be annotated: 435,794,334 (96.1% of unfiltered reads)
   -not annotated reads, aligned in unannotated regions: 72,073,770 (16.5% of mapped reads)
   -annotated reads: 363,720,564 (83.5% of mapped reads)
   -miRNA reads: 363,720,564 (100% of annotated)

The row data were uploaded in the European Nucleotide Archive under the accession number PRJEB24026.

The sequence quality was assessed with FastQC software (www.bioinformatics.babraham.ac.uk/projects/fastqc/).

### Identification of conserved and novel miRNAs by bioinformatic analysis

To obtain clean reads, the adaptator sequences (from the preparation of libraries) and the four nucleotides inserted between the adaptor sequence and the sequence of the miRNAs were removed

using the Cutadapt tool. The *Rattus norgevicus* genome reference was downloaded from UCS Genome Browser assembly ID:m6. All the bioinformatic analysis were performed using the program miRPro (Shi et al, 2015), which uses the main algorithm of the software miRDeep2 (Friedländer et al, 2012). The main advantage of miRPro compared with miRDeep2 is that it proposes consistent and unified names for novel precursors and their mature miRNAs in all libraries of the samples. The reads issued from the sequencing are aligned against the genome of reference using Novoalign (Li & Homer, 2010).

Sequences that matched known rat small RNAs such as rRNA, scRNA, snoRNA, snRNA, and tRNA or degradation fragments of mRNAs were excuded in further analysis, and sequences that perfectly matched the rat genome along their entire lenght and recognized as miRNAs were subjected to subsequent analysis.

The small cleaned RNAs aligned against the miRNA database that cannot be annotated to any previous category were subjected to novel miRNA prediction software miRPro (Shi et al, 2015). Briefly, after the matching of the reads with the genome of reference, the genome is scanned from 5′ to 3′ to excise potential precursor sequences. Then, the software creates an index of potential precursors. Afterwards, the reads are aligned with the potential precursors. Finally, the predictive structures are performed by the software RNAfold.

### Selection of miRNAs

To avoid false-positive miRNAs, we kept miRNAs found at least in two samples of each structure. Then, we kept miRNAs that obtained more than 1 count per million of sequences, in at least two samples per structure.

### Orthologs among new miRNAs discovered

Sequences of new miRNAs were aligned against mouse and human sequences (obtained from miRBase release 21 [http://www.mirbase.org/]) (Kozomara & Griffiths-Jones, 2011, 2014) using Blast software (https://blast.ncbi.nlm.nih.gov/Blast.cgi). Ortholog sequences were determined from a sequence homology more than 95% of identity between the two sequences. Moreover, functional orthologs were determined with a perfect homology into the seed region.

### Characterization of novel miRNA nucleotidic sequences

We used the software suite MEME (http://meme-suite.org) to statistically analyze the nucleotidic composition of novel miRNA sequences.

### Identification of differentially expressed miRNAs and specific miRNAs

We compared the miRNA expression profiles between the nervous system structures, using DESeq2 (Love et al, 2014) under software R. We considered miRNAs as differentially expressed between structures when $P < 0.05$. We considered an miRNA as specific of a structure when it was mainly overexpressed or underexpressed in one structure compared with all the others, following these criteria: log2 fold change > 4 or < −4 and $P < 0.05$.

### miRNA target prediction

To understand the function of selected known miRNAs, based on their abundance, and/or their ability to be differentially expressed, we used the algorithm of the database Targetscan 7.1 (http://www.targetscan.org/vert_71/) (Shin et al, 2010; Agarwal et al, 2015). For novel miRNAs, we used miRDB (http://www.mirdb.org/). We selected mRNA targets that have particular known functions in the nervous system. The signaling pathway of mRNA targets was analyzed using GO Enrichment analysis (http://www.geneontology.org/).

### Preparation and sequencing of mRNA libraries

The libraries were synthesized from 1 μg of RNA for each sample, following the manufacturer's instructions (NEXTflex Illumina Rapid Directional mRNA-Seq, Bioo Scientific Corporation). Briefly, mRNAs were isolated with poly(A) magnetic beads. Then, poly(A) RNA is fragmented, followed by the first and second strand syntheses. 5′ and 3′ adenylated adapters are added, and cDNA is amplified by PCR. Finally, the quality and the quantity of libraries were analyzed with a DNA chip (Agilent) and pooled before sequencing. Sequencing was performed by the Genewiz company on a Illumina HiSeq in the 2 × 150 bp configuration. From the 15 samples, we obtained a total of 408,628,260 reads with a mean quality score of >38.44 and more than 92.48% of bases with a Phred quality score of >30. Reads were aligned with STAR (Dobin et al, 2013), and abundance data (gene counts) were generated with the –quantMode option. Raw mRNA-Seq data and gene count numbers were submitted to the Gene Expression Omnibus database and recorded with the accession number GSE119349.

### Luciferase experiment

The wild-type 3′UTR LCE2D luciferase reporter was obtained by annealing 50-bp synthesized oligonucleotides containing the putative mmu-miR-676 binding site. Mutated 3′UTR construct was obtained using the same 50-bp synthesized oligonucleotides, although the putative mmu-miR-676 binding site composed of 8 nt was replaced with antisense nucleotides. Annealed oligonucleotides were then ligated into pmiRGLO vector (Promega) downstream of the *Firefly* luciferase reporter. The pmiRGLO vector also contains a *Renilla* luciferase cassette that is used as a transfection normalizer. For the assay, LCE2D luciferase reporter in its wild type or mutated form was co-transfected in COS cells with a pcDNA3.1 plasmid expressing the mmu-miR-676, the novel-miR-21, or the Cel-miR-67 as control. Hence, Cel-miR-67 is a miRNA from *C. elegans* known to have no target in mammals. The activity of both *Firefly* and *Renilla* luciferases was assessed 24 h after transfection using the Dual-Luciferase Reporter Assay System kit (Promega). Translation activity was reported as the ratio between *Firefly* and *Renilla* luciferase.

### Statistical analysis

To evaluate the differential expression of miRNAs in the five structures, we used the DESeq2 algorithm in R software. The Shapiro–Wilk test was realized with R on the sequence count results to evaluate the normal distribution of the sample. Then, in regard to

the distribution of the sample, we performed either a Kruskal–Wallis test, followed by a Dunn multiple comparison test, or a one-way analysis of variance, followed by a Bonferroni post hoc test. To evaluate the differential expression of mRNAs in the five structures, we used the DESeq2 algorithm in R software with the Wald test to calculate the $P$-value. To assess the difference in the cumulative frequency distribution of expression change of all miRNA target genes versus nontarget genes, we used the Kolmogorov–Smirnov test. The significance level was set at $P < 0.05$.

# Supplementary Information

# Acknowledgements

A Soula was funded by the French Ministry for Research and Fondation pour la Recherche Médicale. Part of the computations, reads alignment, novel miRNA prediction, and gene abundance estimation were performed at the Bordeaux Bioinformatics Center (CBiB). Computer resources were provided by the CBib, Université de Bordeaux.

## Author Contributions

A Soula: conceptualization, data curation, formal analysis, investigation, methodology, and writing—original draft, review, and editing.
M Valere: data curation, investigation, and methodology.
MJ López-González: investigation and methodology.
V Ury-Thiery: investigation and methodology.
A Groppi: resources, software, methodology, and writing—review and editing.
M Landry: conceptualization, investigation, and methodology.
M Nikolski: conceptualization, resources, software, writing—review and editing.
A Favereaux: conceptualization, formal analysis, funding acquisition, investigation, methodology, project administration, and writing—original draft, review, and editing.

## Conflict of Interest Statement

The authors declare that they have no conflict of interest.

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
