## [Reviewer comments · Life Science Alliance]

Life Science Alliance

Small RNA-Seq reveals novel miRNAs shaping the transcriptomic identity of rat brain structures

Anaïs Soula, Mélissa Valere, María-José López-González, Vicky Ury-Thiery, Alexis Groppi, Marc Landry, Macha Nikolski, and Alexandre Favereaux

DOI: [10.26508/lsa.201800018](https://doi.org/10.26508/lsa.201800018)

Corresponding author(s): Alexandre Favereaux, Bordeaux University

Review Timeline:

Submission Date:	2018-01-09
Editorial Decision:	2018-02-14
Revision Received:	2018-09-04
Editorial Decision:	2018-09-18
Revision Received:	2018-10-10
Accepted:	2018-10-11

Scientific Editor: Andrea Leibfried

Transaction Report:

February 14, 2018

Re: Life Science Alliance manuscript #LSA-2018-00018-T

Dr. Alexandre Favereaux
Bordeaux University
CNRS UMR 5297 Institut Interdisciplinaire de Neurosciences
146 rue Léo Saignat
Bordeaux Cedex 33077
FRANCE

Dear Dr. Favereaux,

Thank you for submitting your manuscript entitled "Small RNA-Seq reveals novel miRNAs shaping the transcriptomic identity of rat brain structures" to Life Science Alliance. The manuscript was assessed by expert reviewers, whose comments are appended to this letter. We invite you to submit a revision if you can address the reviewers' key concerns, as outlined here.

As you will see, the referees think that your work could be a valuable resource if revised. They provide constructive reports and the requested changes seem straightforward to address. However, I would like to stress that it is important to really do so in order to elevate your manuscript to the kind of resource of value to the community that we'd like to publish. Importantly, there are some technical concerns that need to be addressed, and the nomenclature of the identified miRNAs should allow identifying mouse and human orthologues (referee #2, point 1). Furthermore, some functional validation as proof-of-concept should be added. This latter point was raised by both referee #2 and #3, and referee #1 agreed with this view during our cross-commenting session.

-- High-resolution figure, supplementary figure and video files uploaded as individual files: See our detailed guidelines for preparing your production-ready images, <http://life-science-alliance.org/authorguide>

B. MANUSCRIPT ORGANIZATION AND FORMATTING:

Full guidelines are available on our Instructions for Authors page, <http://life-science-alliance.org/authorguide>

Thank you for this interesting contribution to Life Science Alliance. We are looking forward to receiving your revised manuscript.

Sincerely,

Reviewer #1 (Comments to the Authors (Required)):

Soula et al profiled miRNomes of five different brain structures of rat, and discovered structure-specific miRNAs, as well as many novel miRNAs. The story is straightforward, and the analysis is thorough. The information provided by the authors will be of interest for many researchers in neuroscience and RNA biology. However, a few concerns still need to be addressed.

All main points are supported by the data, except the RT-PCR. Additional experiments can be finished within 1 to 2 weeks.

Major comments:

1. Most of authors' analysis is based on fold change. However, low abundant miRNAs often show larger fold changes than highly abundant ones. Therefore, without analyzing, or at least providing, abundance data, the current version of manuscript has not provided enough information to future readers. The authors are highly recommended to provide abundance data (count per million, or similar) for each mature miRNA in each replicate of each structure, either via a supplemental Excel file or depositing to GEO. And in Figure 5, authors can use color code to discriminate miRNAs of different abundance levels, instead of redundantly using green/red to show up- or down- regulation.
2. This reviewer feels difficult to understand the implication of "Chromosomal repartition of pre-miRNAs" section. Especially, what do authors suggest by their finding of enrichment of miRNA genes in positive strand?
3. For RT-PCR correlation between miRNA and targets, multiple experimentally validated mRNAs should be used, at least for miRBase-referenced miRNAs, such as miR-10a or 10b. These well studied miRNAs have many targets validated by luciferase reporter system and CLIP-Seq. Those targets are of more value than predicted targets.

Minor comments:

1. Authors should provide their means to make sure no cross contamination between different brain structures.
2. In the last paragraph of "Chromosomal repartition of pre-miRNAs" section, do the miRNAs locating in different regions have different levels of abundance?
3. What's the difference between Figure 5 and Figure 6? Only the cut-offs of fold change?
4. RT-PCR specificity is a common issue for miRNA quantification because of short sequences. LNA assay and TaqMan assay are currently only ones that have fine specificity. However, miScript assay has done nothing to improve specificity. The authors are recommended to at least discuss such concerns and provide information of their melting curves.
5. In Introduction, authors cited two papers about miR-132. One stated inhibition of plasticity, while the other stated formation of plasticity. As a well-known neuroprotector, miR-132 is more recognized to promote plasticity. The authors are recommended to rephrase this paragraph to be self-consistent.
6. In the second section of Methods, why authors used RNA 6000 Nano (typo of "600" assumingly) which is a total RNA chip, after doing enrichment of the small RNA?
7. In the third section of Methods, please provide the sequencing mode (SE75?). And what "with a depth of 8 millins of reads" means? Also please provide the version number of miRBase used.
8. In Figure 1, for several miRNAs that are specifically detected in only one structure, the miRNA names are suggested to label in the Venn diagram.

9. In Figure 2, the legend is not consistent with the figure.
10. In Figure 4, please provide the information of the x-axis of color bar in the legend.
11. In Figure 12, normalization RNA should be labeled for each panel.
12. In the first section of Results, "size < 15b" should be "size < 15bp"

Reviewer #2 (Comments to the Authors (Required)):

In this manuscript, Soula et al. generated a resource of miRNAs expressed in 5 different regions of the rat central nervous system. It is a resource that is likely to be useful to the scientific community. However, the manuscript is only descriptive and there is limited exploration of the findings. Specific comments:

- The authors use their own nomenclature for the identified miRNAs. It would be useful if they would submit the novel miRNAs to miRbase registry. Also, when miRNAs have human mouse orthologues, it would be advisable to name them according to the human/mouse nomenclature.
- In several figures in the manuscript the authors analyse separately known miRNAs, novel miRNAs and then both together and also describe them separately in the text. This is redundant and repetitive, I would advise the authors to focus in the text and main figures on all miRNAs identified.
- There are several analyses which the relevance is unclear (such as the strand where the miRNAs are located, the position in the genome and the correlation with length), the authors should discuss what might be the novelty. It might be interesting that the X-chromosome and some other chromosomes have a higher number of miRNAs, but if that is the case, the authors should discuss it. Is it really surprising that longer chromosomes have a higher number of miRNAs? Also the r^2 values are not very suggestive of a correlation. More interesting is the exact location of the miRNAs, it would be good to add some genome browser screenshots to complement figure 2G.
- The authors refer that the majority of the novel miRNAs identified do not have homologues in mouse and human. The authors should discuss this in more detail. Is it because there are no similar studies with small RNA-Seq in different regions of the brain in mouse and human? Or could they reflect species-specific miRNAs, with species-specific functions, which would be quite interesting?
- The authors are sometimes not very precise in the text. They state for instance that "heatmap representation (...) showed (...) near-perfect clustering". What is near-perfect clustering? It is not the heatmap representation that results in the clustering, the authors performed hierarchical clustering rather? That should be clear in the text. The heatmap is just a visualization, not a clustering tool. They also state "clustering coefficient is better(...)". This is a subjective statement.
- The authors use Circle plots to illustrate putative mRNA targets and gene ontology, but some of them are too convoluted and impossible to read (Figure 7, for instance). Tables and Barplots would

be a better way to display the dataset, so the readers can easily explore the dataset.

- The proof of concept experiments are very limited. The authors present a small subset of putative targets of the miRNAs and show anticorrelation of expression. It is not necessary that this anticorrelation of expression is direct and other targets might not show the same anticorrelation. The authors should validate a much larger number of targets. Comparison with mRNA-Seq data from the exact same regions would be ideal, and would greatly increase the value of this resource.

- The authors overstate the relevance of their data in some occasions in the manuscript. For example "These results suggest the critical role played by these enriched/depleted miRNAs in the transcriptomic and functional identities of CNS structures" is over the top. The authors need to tone down their statements.

Reviewer #3 (Comments to the Authors (Required)):

Here authors employed small RNA-Seq technology to analyze miRNomes of rat olfactory bulb, the cortex, the hippocampus, the striatum and the spinal cord. Out of the 495 miRNAs referenced in miRBase, 365 were found expressed in these five CNS structures and many of these showed CNS structure-specific expression of distinct miRNAs. The authors also claim to have discovered a hundred of novel miRNAs with the orthologous present in mouse or human. Using target prediction of these different miRNomes and Gene Ontology, they claim that these miRNAs may play important role in CNS transcriptome regulation. While it is interesting to have the miRNome of the rat CNS, a number of concerns should be addressed before the manuscript is published:

-Other studies have used small RNA-Seq to study the miRNome of distinct part the rat brain in adults or on different stages of development. Authors should provide comprehensive comparisons with these studies and highlight novel aspects that is revealed by their study.

-The study employs miRNA prediction algorithms, which come with their own weaknesses. The authors should comment on what this could mean for their interpretations and suggest precautions.

-Authors should also explore whether they find any specific features in miRNAs and their target mRNAs from brain versus those from other rat tissues.

-Importantly, the authors do not provide any wet lab or functional validation of their findings. This is a key weakness of this manuscript. They should include some in situ hybridization of structure-specific miRNAs and if possible, manipulation of some relevant miRNAs combined with studying changes in target mRNA.

-They found some miRNAs that they refer to as "functional orthologs", meaning that they share the same seed region with other miRNAs. To test whether they are functionally similar, further analysis should be performed for at least a few such siRNAs e.g. using reporter-gene assays.

Reviewer #1 (Comments to the Authors (Required)):

Soula et al profiled miRNomes of five different brain structures of rat, and discovered structure-specific miRNAs, as well as many novel miRNAs. The story is straightforward, and the analysis is thorough. The information provided by the authors will be of interest for many researchers in neuroscience and RNA biology. However, a few concerns still need to be addressed.

All main points are supported by the data, except the RT-PCR. Additional experiments can be finished within 1 to 2 weeks.

Major comments:

1. Most of authors' analysis is based on fold change. However, low abundant miRNAs often show larger fold changes than highly abundant ones. Therefore, without analyzing, or at least providing, abundance data, the current version of manuscript has not provided enough information to future readers. The authors are highly recommended to provide abundance data (count per million, or similar) for each mature miRNA in each replicate of each structure, either via a supplemental Excel file or depositing to GEO. And in Figure 5, authors can use color code to discriminate miRNAs of different abundance levels, instead of redundantly using green/red to show up- or down- regulation.

We agree with reviewer that abundance data are important. Actually, abundance data were present in the first version of the manuscript since the heatmaps were colour-coded according to the count per million of miRNA. We agree that this information is not easy to read and we now provide abundance data concerning known and novel miRNAs as a supplementary table of count per million (Supp. Table 7). Concerning the new mRNA-Seq data they were deposited to GEO and the accession number is GSE119349.

2. This reviewer feels difficult to understand the implication of "Chromosomal repartition of pre-miRNAs" section. Especially, what do authors suggest by their finding of enrichment of miRNA genes in positive strand?

We are sorry that this section was not clear, we performed this series of analyzes to try to understand why these novel miRNAs were not discovered before and to look for new miRNA clusters. While we did not find any new cluster, we found that the number of miRNA located on the positive strand is higher than the one on the negative strand. This is true for both known and novel miRNAs. These are only descriptive data and to our knowledge there is no clear explanation for the localization of miRNA genes on the plus versus minus strand of the genomic DNA. We added these explanations to the result section to help readers.

3. For RT-PCR correlation between miRNA and targets, multiple experimentally validated mRNAs should be used, at least for miRBase-referenced miRNAs, such as miR-10a or 10b. These well studied miRNAs have many targets validated by luciferase reporter system and CLIP-Seq. Those targets are of more value than predicted targets.

We agree with reviewer and we have removed RT-qPCR data. We have now performed a mRNA-Seq analysis of the very same samples, thus looking at all mRNAs and not only a selection of targets. Then, we used target prediction to correlate miRNA and mRNA expressions. The results of this unbiased pan-transcriptomic analysis shows that many predicted targets of specific miRNAs are actually regulated. In addition, GO term analysis and review of the literature strongly suggest that these specific miRNAs (that is miRNAs highly abundant or highly

depleted in a structure) and their regulated targets have an impact on the neurobiological function of these brain structures.

Minor comments:

1. Authors should provide their means to make sure no cross contamination between different brain structures.

Cross contamination is a key issue in transcriptomic studies. To avoid cross contaminations, we selected brain structures that are easy to locate and dissect. Besides, we used new disposable dissecting instruments for each sample. In addition, a common source of contamination is blood which contain many RNAs (either cellular or cell-free). To avoid contamination from blood, we performed perfusion with ice cold artificial cerebro-spinal fluid before dissecting brain areas. These details have been added to the material and method section.

2. In the last paragraph of "Chromosomal repartition of pre-miRNAs" section, do the miRNAs locating in different regions have different levels of abundance?

The level of expression of miRNAs was not correlated to their position on the genome (data not shown).

3. What's the difference between Figure 5 and Figure 6? Only the cut-offs of fold change?

We agree with reviewer that the difference between Figure 5 and Figure 6 was not clear enough. Figure 5 is the list of all the miRNAs that are differentially expressed in the CNS structures according to the statistical analysis. However, this list contains miRNAs with a small, but statistically significant, difference in expression between structures. To focus on the most biologically relevant miRNAs, we added a cut-off in the fold-change to produce a short list of specifically enriched or depleted miRNAs : miRNAs listed in the Figure 6. We now clarify this point in the manuscript.

4. RT-PCR specificity is a common issue for miRNA quantification because of short sequences. LNA assay and TaqMan assay are currently only ones that have fine specificity. However, miScript assay has done nothing to improve specificity. The authors are recommended to at least discuss such concerns and provide information of their melting curves.

We agree with reviewer that RT-qPCR of miRNAs is not trivial and that LNA and Taqman assay have higher specificity than SYBR green methods. However, these techniques are very expensive and thus not adapted to screening strategies where you want to analyze dozens if not hundreds of miRNA species. To overcome this issue we slightly modified the design of our miRNA primers in accordance to the NCODE VILO kit from Invitrogen. Briefly, to increase the specificity of miRNA primers when the GC content of the mature miRNA is low we added extra G and C nucleotides on the 5' end of the primer. This strategy increases the T_m of the primer and thus the specificity of the quantification. Finally to confirm the validity of RT-qPCR amplification a melting curve was performed at the end of each qPCR run and any suspicious data were removed.

However, since we have now performed mRNA-seq analysis on the very same samples used for miRNA analysis and made a correlation between miRNA and mRNA expression, we have removed the RT-qPCR data that were obsolete.

5. In Introduction, authors cited two papers about miR-132. One stated inhibition of plasticity, while the other stated formation of plasticity. As a well-known neuroprotector,

miR-132 is more recognized to promote plasticity. The authors are recommended to rephrase this paragraph to be self-consistent.

Done

6. In the second section of Methods, why authors used RNA 6000 Nano (typo of "600" assumingly) which is a total RNA chip, after doing enrichment of the small RNA?

We are sorry for the typo about the Agilent RNA 6000 Nano chip. In fact the miRNeasy Micro kit that we used for RNA extraction does not enrich in small RNA but rather enables the isolation of all RNAs, including the small RNAs. The isolation protocol has been corrected too. We used this strategy to be able to perform both small RNA and mRNA sequencing on the very same samples. In this revised version of the manuscript we actually perform mRNA sequencing.

7. In the third section of Methods, please provide the sequencing mode (SE75?). And what "with a depth of 8 millions of reads" means? Also please provide the version number of miRBase used.

Here is the requested information : The next generation single-read sequencing was performed using the sequencer NextSeq 500, Illumina, using the single end 75bp high output sequencing mode. We selected this mode to achieve a minimum of 8 millions of reads per sample to enable accurate detection of novel miRNAs with the miRPro algorithm. Release 21 of miRBase was used for analyses, it is now indicated in the material and method section.

8. In Figure 1, for several miRNAs that are specifically detected in only one structure, the miRNA names are suggested to label in the Venn diagram.

This comment is in line with reviewer 2 that we « analyse separately known miRNAs, novel miRNAs and then both together and also describe them separately in the text. This is redundant and repetitive ». So in this new version of the manuscript we removed two out of the three Venn diagrams.

9. In Figure 2, the legend is not consistent with the figure.

Sorry for the error, the legend has been corrected.

10. In Figure 4, please provide the information of the x-axis of color bar in the legend.

Done

11. In Figure 12, normalization RNA should be labeled for each panel.

This figure has been removed since we replaced RT-qPCR data with mRNA-Seq data.

12. In the first section of Results, "size < 15b" should be "size < 15bp"

Done

Reviewer #2 (Comments to the Authors (Required)):

In this manuscript, Soula et al. generated a resource of miRNAs expressed in 5 different regions of the rat central nervous system. It is a resource that is likely to be useful to the scientific community. However, the manuscript is only descriptive and there is limited exploration of the findings. Specific comments:

- *The authors use their own nomenclature for the identified miRNAs. It would be useful if*

they would submit the novel miRNAs to miRbase registry. Also, when miRNAs have human mouse orthologs, it would be advisable to name them according to the human/mouse nomenclature.

We agree with reviewer that miRNA nomenclature is key point in miRNA research. However, submission of new miRNA sequences to miRBase must be performed after manuscript acceptance accordingly to miRBase submission guidelines (<http://www.mirbase.org/help/submit.shtml>). Final names will be assigned on acceptance. We agree with reviewer, concerning orthologs we will ask miRBase to give them a name according to human/mouse nomenclature.

- In several figures in the manuscript the authors analyse separately known miRNAs, novel miRNAs and then both together and also describe them separately in the text. This is redundant and repetitive, I would advise the authors to focus in the text and main figures on all miRNAs identified.

We agree with reviewer and in this revised version of the manuscript we removed, as much as we could, the redundant analysis of known, novel and known+novel miRNAs.

- There are several analyses which the relevance is unclear (such as the strand where the miRNAs are located, the position in the genome and the correlation with length), the authors should discuss what might be the novelty. It might be interesting that the X-chromosome and some other chromosomes have a higher number of miRNAs, but if that is the case, the authors should discuss it. Is it really surprising that longer chromosomes have a higher number of miRNAs? Also the r2 values are not very suggestive of a correlation. More interesting is the exact location of the miRNAs, it would be good to add some genome browser screenshots to complement figure 2G.

This concern was also raised by reviewer #1. We are sorry that this section was not clear, we performed this series of analyzes to try to understand why these novel miRNAs were not discovered before and to look for new miRNA clusters. While we did not find any new cluster, we found that the number of miRNA located on the positive strand is higher than the one on the negative strand. This is true for both known and novel miRNAs. These are only descriptive data and to our knowledge there is no clear explanation for localization of miRNA genes on the plus versus minus strand of the genomic DNA. We added these explanations to the result section to help readers.

As suggested by reviewer, we added genome browser screenshots to complement figure 2G.

- The authors refer that the majority of the novel miRNAs identify do not have homologues in mouse and human. The authors should discuss this in more detail. Is it because there are no similar studies with small RNA-Seq in different regions of the brain in mouse and human? Or could they reflect species-specific miRNAs, with species-specific functions, which would be quite interesting?

This is an interesting question but it is really hard to give a firm answer. Since mouse and human brains had already been analyzed for new miRNAs, it is likely that these miRNAs are rat-specific, supporting rat-specific gene regulations. Studies on human genome suggest that recently discovered miRNAs are mainly evolutionarily young. In addition, it has been recently shown that within a species, interstrain variations exist with functional significance on the targeted mRNAs. However, we cannot exclude that the mouse and human orthologs of these miRNAs have not been found yet due to the bioinformatic tools used for miRNA prediction.

In conclusion, more experiments in mouse and human are needed to reach a valuable conclusion. We added these comments to the discussion section.

- The authors are sometimes not very precise in the text. They state for instance that "heatmap representation (...) showed (...) near-perfect clustering". What is near-perfect clustering? It is not the heatmap representation that results in the clustering, the authors performed hierarchical clustering rather? That should be clear in the text. The heatmap is just a visualization, not a clustering tool. They also state "clustering coefficient is better(...)". This is a subjective statement.

Sorry for the lack of precision in this part of the manuscript. We have now modified the text to clearly indicate that « To test if miRNA expression discriminated the different brain structures, we performed hierarchical clustering analysis of the miRNome of the five CNS structures tested. To perform this analysis, we used the count per million value for each miRNA (Supp. Table 7) and we represented the results as a heatmap. »

- The authors use Circle plots to illustrate putative mRNA targets and gene ontology, but some of them are too convoluted and impossible to read (Figure 7, for instance). Tables and Barplots would be a better way to display the dataset, so the readers can easily explore the dataset.

We agree with reviewer, in the new version of the manuscript we removed circos plot and replace them with barplots.

- The proof of concept experiments are very limited. The authors present a small subset of putative targets of the miRNAs and show anticorrelation of expression. It is not necessary that this anticorrelation of expression is direct and other targets might not show the same anticorrelation. The authors should validate a much larger number of targets. Comparison with mRNA-Seq data from the exact same regions would be ideal, and would greatly increase the value of this resource.

We completely agree with reviewer and we now have performed a mRNA-Seq analysis of the very same samples. Then, we used target prediction to correlate miRNA and mRNA expression. These results strongly suggest that specific miRNA (that is miRNAs highly abundant or highly depleted in a structure) have an impact on the expression of their mRNA targets.

- The authors overstate the relevance of their data in some occasions in the manuscript. For example "These results suggest the critical role played by these enriched/depleted miRNAs in the transcriptomic and functional identities of CNS structures" is over the top. The authors need to tone down their statements.

We agree with reviewer and accordingly toned down our statements and concomitantly added new experiments to support our conclusions.

Reviewer #3 (Comments to the Authors (Required)):

Here authors employed small RNA-Seq technology to analyze miRNome of rat olfactory bulb, the cortex, the hippocampus, the striatum and the spinal cord. Out of the 495 miRNAs referenced in miRBase, 365 were found expressed in these five CNS structures and many of these showed CNS structure-specific expression of distinct miRNAs. The authors also claim to have discovered a hundred of novel miRNAs with the orthologous present in mouse or human. Using target prediction of these different miRNomes and

Gene Ontology, they claim that these miRNAs may play important role in CNS transcriptome regulation. While it is interesting to have the miRNome of the rat CNS, a number of concerns should be addressed before the manuscript is published:

-Other studies have used small RNA-Seq to study the miRNome of distinct part the rat brain in adults or on different stages of development. Authors should provide comprehensive comparisons with these studies and highlight novel aspects that is revealed by their study.

We agree with reviewer, in the discussion section we now provide a comprehensive comparison with previous studies and we now highlight the novelty of our study. Thus, we compare our novel miRNAs those previously found by Yin *et al.* In addition, to our knowledge, we are the first ones to identify specifically enriched/depleted miRNAs in the different structures of the rat brain and to link these results with transcriptomic data thereby identifying miRNAs that may have a crucial role in the specification of CNS structures.

-The study employs miRNA prediction algorithms, which come with their own weaknesses. The authors should comment on what this could mean for their interpretations and suggest precautions.

We agree that miRNA prediction algorithms are not perfect, and could lead to misleading results. In the discussion we added a comment on that and suggested a strategy to limit the number of false positive: “A strategy to limit the number of false positive novel miRNAs and to focus and the most biologically relevant ones could be to apply a filter on the predicted novel miRNAs based on the abundance. Thus, we initially obtained more than 8000 novel miRNAs from our samples with the miRPro algorithm (based on miRDeep2) and we reduced this number to 90 by applying two filters : the number of reads (>1cpm of total reads), and the presence of the novel miRNA in at least two independent samples.”

-Authors should also explore whether they find any specific features in miRNAs and their target mRNAs from brain versus those from other rat tissues.

Concerning known miRNAs enriched/depleted in the CNS structures, nucleotidic composition analysis did not show any difference compared to all miRNAs in the rat miRBase (data not shown). Concerning novel miRNAs, they did not show any specific nucleic composition (Figure 3A). Finally, concerning the target mRNA, we ran a Gene Ontology term enrichment analysis that clearly show a bias towards neurobiological mechanisms (Figures 7C, 8B, 9B, 10C). This latter result strongly suggests the relevance of the identified targets.

-Importantly, the authors do not provide any wet lab or functional validation of their findings. This is a key weakness of this manuscript. They should include some in situ hybridization of structure-specific miRNAs and if possible, manipulation of some relevant miRNAs combined with studying changes in target mRNA.

We completely agree with reviewer and we have now performed a mRNA-Seq analysis of the very same samples. Then, we used target prediction to correlate miRNA and mRNA expression. These results strongly suggest that specific miRNA (that is miRNAs highly enriched or depleted in a structure) have an impact on the expression of their mRNA targets.

-They found some miRNAs that they refer to as "functional orthologs", meaning that they share the same seed region with other miRNAs. To test whether they are functionally

similar, further analysis should be performed for at least a few such siRNAs e.g. using reporter-gene assays.

We agree with reviewer that the concept of “functional orthologs“ needs to be tested experimentally. To evaluate the ability of these “functional orthologs” to repress mRNA expression the same way as miRNAs sharing the same seed region, we designed a proof-of-concept experiment based on a luciferase assay. Novel-rno-miR-21-5p was predicted to be a “functional ortholog” of mouse mmu-miR-676-3p and thus may regulate the same target genes. We searched Targetscan for the predicted targets of mmu-miR-676-3p and from the top five predicted targets, we selected LCE2D. LCE2D was selected because the interaction with mmu-miR-676-3p relied only on the hybridization of the seed region and interestingly, novel-rno-miR-21-5p was also predicted to interact with LCE2D only via the seed region. We cloned the 3'UTR sequence of LCE2D in a luciferase reporter plasmid and tested in COS cells the ability of novel-rno-miR-21-5p to regulate LCE2D. Figure 3B shows that mmu-miR-676-3p inhibited the expression of the luciferase reporter by 15% compared to control condition and more interestingly, novel-rno-miR-21-5p inhibited reporter-gene by 22%. The level of regulation was moderate but this is a common feature of miRNAs which are known as fine-tuner of gene expression. The key point is that both miRNAs were able to regulate the same target and this can be considered as a clue that “functional orthologs” may regulate the same targets.

September 18, 2018

RE: Life Science Alliance Manuscript #LSA-2018-00018-TR

Dr. Alexandre Favereaux
Bordeaux University
CNRS UMR 5297 Institut Interdisciplinaire de Neurosciences
146 rue Léo Saignat
Bordeaux Cedex 33077
France

Dear Dr. Favereaux,

Thank you for submitting your revised manuscript entitled "Small RNA-Seq reveals novel miRNAs shaping the transcriptomic identity of rat brain structures". As you will see, the reviewers appreciate the introduced changes, and we would be happy to publish your paper in Life Science Alliance pending final revisions necessary to address reviewer #2's concerns on the new data and to meet our formatting guidelines.

The points raised by previous reviewer #2 all seem valid and straightforward to address by minor changes using the data already at hand / including text changes. We would like to ask you furthermore to mention Fig. 9A in the manuscript text, and to provide the manuscript text in a docx file that also includes the figure legends. All figures should be provided as individual files and without legend, please.

A. FINAL FILES:

-- High-resolution figure, supplementary figure and video files uploaded as individual files: See our detailed guidelines for preparing your production-ready images, <http://life-science-alliance.org/authorguide>

-- Summary blurb (enter in submission system): A short text summarizing in a single sentence the study (max. 200 characters including spaces). This text is used in conjunction with the titles of papers, hence should be informative and complementary to the title. It should describe the context and significance of the findings for a general readership; it should be written in the present tense

and refer to the work in the third person. Author names should not be mentioned.

B. MANUSCRIPT ORGANIZATION AND FORMATTING:

Full guidelines are available on our Instructions for Authors page, <http://life-science-alliance.org/authorguide>

Thank you for your attention to these final processing requirements.

Sincerely,

Reviewer #2 (Comments to the Authors (Required)):

In the revised version of their manuscript, Soula et al. address my major criticisms, and importantly perform mRNA-Seq on the same samples previously analysed, correlating mRNA expression with

the miRNA profile. This significantly strengthens the manuscript as a resource, and I now deem that the manuscript can be accepted for publication in LSA, pending the following points:

- In the mRNA-Seq analysis, the authors focus on the mRNAs predicted to be targeted by the differentially expressed miRNAs and that indeed show correlating expressions. However, it is not clear how were target genes displayed in figures 7 and 10 selected. The authors should analyse in an unbiased way what is the actual percentage of all predicted target genes that are differentially expressed. Are there many genes predicted to be regulated that do not present correlative expression? Is there any "regulation" in the opposite direction then expected? The authors should represent this as a shift in cumulative frequency distribution of expression change of all miRNA target genes vs non-targeted genes. This is important information that must be included in the paper, since it might shed some light on the accuracy/relevance of predictions of miRNA targeting.
- While I understand the relevance of trying to determine if the miRNAs identified are clustered in the genome, I still consider that the analysis of correlation between length of chromosome/strand and number of miRNAs, and general location in the chromosome, have too much emphasis in the manuscript. I would advise to significantly shorten the corresponding paragraphs or move to discussion.
- The authors include a new luciferase experiment as a proof of concept of functional orthology between rat and mouse miRNAs. The effects observed are not very strong and the authors do not include enough detail on the experiment:
 - o there is no description in the Methods of the experiment
 - o what is control, a non targeting miRNA?
 - o how many times was the experiment repeated, the standard errors(SEM, SD?) represent technical or biological errors?
 - o The author appear to normalize all experiments to control. If so, is 1-away Anova the appropriate statistical test, given that the variation of control will be non-existent?
- The text in the x-axis in the heatmaps in Figure 4 is too small, the authors should add the name of the differential expressed miRNAs next to the arrow heads in larger font
- Heatmaps in figure 4 are redundant and either novel and known heatmaps, or novel+known in one heatmap should be shown
- Each supplementary tables should have a title, a descriptive legend and a legend describing what each column refers to
- Supplementary Table 9 should include the gene names
- Figure legends need to be better described

Reviewer #3 (Comments to the Authors (Required)):

The authors have sufficiently addressed most of my concerns in the revised version. I recommend publication.

Reviewer #2 (Comments to the Authors (Required)):

In the revised version of their manuscript, Soula et al. address my major criticisms, and importantly perform mRNA-Seq on the same samples previously analysed, correlating mRNA expression with the miRNA profile. This significantly strengthens the manuscript as a resource, and I now deem that the manuscript can be accepted for publication in LSA, pending the following points:

- In the mRNA-Seq analysis, the authors focus on the mRNAs predicted to be targeted by the differentially expressed miRNAs and that indeed show correlating expressions. However, it is not clear how were target genes displayed in figures 7 and 10 selected. The authors should analyse in an unbiased way what is the actual percentage of all predicted target genes that are differentially expressed. Are there many genes predicted to be regulated that do not present correlative expression? Is there any "regulation" in the opposite direction then expected? The authors should represent this as a shift in cumulative frequency distribution of expression change of all miRNA target genes vs non-targeted genes. This is important information that must be included in the paper, since it might shed some light on the accuracy/relevance of predictions of miRNA targeting.

As suggested by reviewer, we ran an analysis on the cumulative frequency distribution of expression change of all miRNA target genes versus non-targeted genes. This analysis showed that the expression of predicted targets is significantly different from the one of non-targeted genes. While in most cases the expression of the predicted targets was regulated as expected (opposite to miRNA expression since miRNAs act as inhibitors), on some occasion the predicted targets showed an unexpected regulation. This result is not surprising since miRNA are not the only regulators of gene expression. Hence, other mechanisms, such as transcription factors, affect mRNA levels and the net effect on mRNA expression is a combination of all these mechanisms. Then, in an attempt to highlight the most biologically relevant targets for each structure, we selected the mRNAs displaying the most important regulation (either the 20 uppest or 20 lowest regulated mRNAs). We now include in the manuscript, for each structure, the cumulative frequency distribution of expression change of all miRNA target genes versus non-targeted genes. We now clearly state the exact number of predicted targets that are actually regulated and the rationale for gene selection.

- While I understand the relevance of trying to determine if the miRNAs identified are clustered in the genome, I still consider that the analysis of correlation between length of chromosome/strand and number of miRNAs, and general location in the chromosome, have too much emphasis in the manuscript. I would advise to significantly shorten the corresponding paragraphs or move to discussion.

In accordance with reviewer's comment, we significantly reduced the result section related to the chromosomal repartition of miRNA genes.

- The authors include a new luciferase experiment as a proof of concept of functional orthology between rat and mouse miRNAs. The effects observed are not very strong and the authors do not include enough detail on the experiment:

o there is no description in the Methods of the experiment

Sorry for the omission, we now provide a full description of luciferase assay in the method section.

o what is control, a non targeting miRNA?

As a control, we used a miRNA from *C. elegans*, Cel-miR-67, known to have no target in mammals. This control miRNA has been previously used in numerous studies. It is now clearly stated in the method and result sections and labeled on the corresponding graph.

o how many times was the experiment repeated, the standard errors(SEM, SD?) represent technical or biological errors?

The luciferase assay was performed on a cell line enabling the production of numerous biological replicates. Thus, the results come from 16, 19 and 18 biological replicates for the three conditions tested : control miRNA (Cel-miR-67), miR-676 and novel-miR-21, respectively. Error bars represent SD of biological replicates.

o The author appear to normalize all experiments to control. If so, is 1-away Anova the appropriate statistical test, given that the variation of control will be non-existent?

Sorry for the lack of precision, it should be more clear now with the luciferase method description. In fact luciferase experiment relies on the expression of a plasmid designed to express both the renilla luciferase under the regulation of the 3'UTR of interest (here LCE2D) and a firefly luciferase that is constitutively express. Thus, renilla luciferase is used to test the regulation exerted by the miRNA on the 3'UTR while the firefly luciferase is used as a normalizer to control transfection efficiency. The luciferase measurement is a ratio between the renilla and the firefly luciferase values. The statistical analysis is performed on this ratio and thus there is variation of this ratio in the control group. As a conclusion, one-way ANOVA is the right test to assess the difference between the control, the mmu-miR-676 and the novel-miR-21 groups. To help readers, luciferase assay data are commonly displayed as normalized values to the control group. To perform this normalization each replicate is normalized to the mean of control values. Thus, even in the control group, the variability between replicates is shown. The biological error is usually very low in these experiments since luciferase constructs are robustly expressed in cell line and since the technical simplicity of this experiment enables the production of numerous biological replicates.

- The text in the x-axis in the heatmaps in Figure 4 is too small, the authors should add the name of the differential expressed miRNAs next to the arrow heads in larger font

- Heatmaps in figure 4 are redundant and either novel and known heatmaps, or novel+known in one heatmap should be shown

As suggested by reviewer we remove th redundant heatmap (novel+known) from figure 4 and we added the name of differentially expressed miRNAs in a larger font. Corresponding text in the result section has been updated.

- Each supplementary tables should have a title, a descriptive legend and a legend describing what each column refers to

Done

- Supplementary Table 9 should include the gene names

Done

- Figure legends need to be better described

Done

Reviewer #3 (Comments to the Authors (Required)):

The authors have sufficiently addressed most of my concerns in the revised version. I recommend publication.

October 11, 2018

RE: Life Science Alliance Manuscript #LSA-2018-00018-TRR

Dr. Alexandre Favereaux
Bordeaux University
CNRS UMR 5297 Institut Interdisciplinaire de Neurosciences
146 rue Léo Saignat
Bordeaux Cedex 33077
France

Dear Dr. Favereaux,

Thank you for submitting your Resource entitled "Small RNA-Seq reveals novel miRNAs shaping the transcriptomic identity of rat brain structures". I appreciate the introduced changes and it is a pleasure to let you know that your manuscript is now accepted for publication in Life Science Alliance. Congratulations on this interesting work.

The final published version of your manuscript will be deposited by us to PubMed Central (PMC) as soon as we are allowed to do so, the application for PMC indexing has been filed. You may be eligible to also deposit your Life Science Alliance article in PMC or PMC Europe yourself, which will then allow others to find out about your work by Pubmed searches right away. Such author-initiated deposition is possible/mandated for work funded by eg NIH, HHMI, ERC, MRC, Cancer Research UK, Telethon, EMBL.

Please also see:

<https://www.ncbi.nlm.nih.gov/pmc/about/authorms/>

<https://europepmc.org/Help#howsubsmanu>

*****IMPORTANT:** If you will be unreachable at any time, please provide us with the email address of an alternate author. Failure to respond to routine queries may lead to unavoidable delays in publication.*******

DISTRIBUTION OF MATERIALS:

Authors are required to distribute freely any materials used in experiments published in Life Science Alliance. Authors are encouraged to deposit materials used in their studies to the appropriate

repositories for distribution to researchers.

Again, congratulations on a very nice paper. I hope you found the review process to be constructive and are pleased with how the manuscript was handled editorially. We look forward to future exciting submissions from your lab.

Sincerely,
